# When unlearning is free: leveraging low influence points to reduce computational costs

## Abstract

As concerns around data privacy in machine learning grow, the ability to *unlearn*—or remove— specific data points from trained models becomes increasingly important. While state-of-the-art unlearning methods have emerged in response, they typically treat all points in the forget set equally. In this work, we challenge this approach by asking: do points that have a negligible impact on the model's learning need to be removed? Through a comparative analysis of influence functions across language and vision tasks, we identify subsets of training data with negligible impact on model outputs. Leveraging this insight, we propose an efficient unlearning framework that reduces the size of datasets before unlearning—leading to significant computational savings (up to ∼50%) on real-world empirical examples.

## 1 Introduction

As machine learning becomes more embedded in user applications, the importance of the data quality used for training grows significantly. However, collecting diverse datasets introduces challenges, including privacy concerns, evolving regulatory requirements, and disputes over data ownership that may result in requests for data removal (Zhang et al., 2023). Consequently, legal, ethical, and business motivations fuel a growing interest in effectively *removing* specific data points from the training dataset of an existing model. This process of selectively removing data from a model is known as *unlearning* (Cao & Yang, 2015).

For a (random) training algorithm $\mathcal{A}$ that trains on dataset $D$, let $S \subset D$ denote the *forget set*—a subset of data items that should be unlearned. Ideally, unlearning produces a model that behaves as if it has never seen $S$. That is, a model that is statistically indistinguishable from a random model $\mathcal{A}(D \backslash S)$ trained without access to $S$. This can be achieved by retraining a model on $D \setminus S$ from an initial random state. However, full retraining is often computationally expensive, especially when unlearning requests are frequent. The objective of an unlearning algorithm, therefore, is to approximate this ideal efficiently by producing a model 'similar' to one drawn from $\mathcal{A}(D \setminus S)$ (Triantafillou et al., 2024).

Unlearning methods that aim to reduce computational cost typically do so by continuing to train an existing model or by directly modifying its parameters (Section 2.0.1). The costs of these methods tend to scale with the sizes of both $D$ and $S$. That the cost scales with the size of the forget set may reflect that a similar effort is spent on unlearning each observation in $S$. However, if some points in $S$ did not contribute substantially to the training of the initial model, then there may be scope to improve efficiency by allocating more compute to unlearning only more influential data-points.

In this work, we investigate whether it is possible to reduce the size of $S$ prior to unlearning without compromising the privacy guarantees of unlearning the full set $S$. Furthermore, we explore whether this reduction also leads to significant decreases in computational costs. We begin by demonstrating that, across datasets, large subsets of low-impact training points exist and can serve as candidates for removal prior to unlearning. To efficiently identify these points, we conduct a comparative analysis of influence-based methods that estimate the importance of individual training samples. The influence of example $(x_i, y_i)$ over example $(x_j, y_j)$ is the average difference between prediction accuracy when the model is trained using the

whole data, and when it is trained on all points except $(x_i, y_i)$. Formally, this is defined as:

$$\Pr_{h \leftarrow \mathcal{A}(D)}[h(x_j) = y_j] - \Pr_{h \leftarrow \mathcal{A}(D \setminus \{x_i\})}[h(x_j) = y_j] \tag{1.1}$$

Our hypothesis is that points that have low influence over all other points in the test set are promising candidates to remove *without additional retraining*, because removing them is unlikely to affect model performance on other points. A special case is the self influence of a point, also known as the *label memorization score* (Feldman, 2021).

For data point $(x_i, y_i)$, $\mathrm{mem}(\mathcal{A}, D, i)$ is defined as

$$\Pr_{h \leftarrow \mathcal{A}(D)}[h(x_i) = y_i] - \Pr_{h \leftarrow \mathcal{A}(D \setminus \{x_i\})}[h(x_i) = y_i] \tag{1.2}$$

Intuitively, points with low self-influence may be well-suited for removal, as they likely have low impact on other points as well. As such, we explore the use of both test and self-influence approximations to identify low-impact points. We provide theoretical motivations for this approach using ideas from (Broderick et al., 2023) who use approximate influence to construct a robustness metric in the context of parameter estimation. The concept of filtering points prior to unlearning, referred to as *unnecessary unlearning* in recent work (Li et al., 2025), has also been explored using similarity-based methods. In contrast, our approach employs influence approximations for filtering, which outperform cosine similarity across our evaluation metrics (Appendix B and Figure B.3).

We next empirically demonstrate that removing these points from the forget set does not compromise the privacy guarantees of original unlearning. Consistent with the literature (Li et al., 2025; Triantafillou et al., 2024), we use four key metrics to measure performance and privacy: execution time, changes in the membership inference attack (MIA) accuracy, accuracy on the forget and retain sets of our unlearned models, as well as, performance on the evaluation metric from the NeurIPS'23 competition on unlearning[1]. Additionally, we also confirm that models retrained without these low-influence points generalize well to them, indicating minimal contribution to learning at all. Together, these results suggest that such points can be safely excluded from the forget set without affecting privacy or model performance. Finally, we leverage our findings to substantially reduce the computational cost of state-of-the-art unlearning methods in real-world settings. We introduce an algorithm-agnostic unlearning framework that uses influence estimates to reduce unlearning datasets prior to execution. We then demonstrate the effectiveness of our framework by integrating it with leading unlearning methods from the NeurIPS'23 competition on unlearning, and testing it on three unlearning scenarios: sample-wise, class-wise and subclass-wise unlearning. Our framework achieves significant reductions in execution time (up to ∼50%) while maintaining privacy and performance comparable to unlearning over the full data across both vision and language tasks.

In summary, our contributions are as follows:

- We compare efficient, influence-based methods for finding low-impact training points $D_{LI} \subset D$ that could act as removal candidates for future unlearning (Sections 3 and 4).

- We show that sets of these points can be safely removed without compromising privacy or performance. (Sections 4 and 5)

- We introduce an algorithm-agnostic framework that reduces unlearning set sizes using influence approximations (Section 5).

- We show how using our unlearning framework across settings can significantly decrease computational costs (Section 5).

---

[1]https://unlearning-challenge.github.io/

## 2 Related Works

### 2.0.1 Unlearning

The unlearning literature often highlights a trade-off between provable guarantees and computational efficiency. One line of work addresses this by developing methods for exact and provable unlearning (Liu, 2024). Bourtoule et al. (2020) introduce SISA where training data is divided into shards with each point appearing only once. In the case of unlearning a point, this implies that only a single model needs to be retrained. While this method can provably unlearn a point, it may be costly to train as models scale.
In contrast, many approximate unlearning algorithms offer cheaper solutions. Top submissions from the NeurIPS'23 competition on unlearning implement methods that focus on manipulating the parameters of a trained model directly (Triantafillou et al., 2024). These submissions either reinitialize a subset of the model's layers (Amnesiacs, Sun, Forget, Kookmin, Sebastian) or apply Gaussian noise to parameters (Seif, Sun). Many methods then perform additional training to improve performance on a retain set. As such, these methods offer a promising alternative to full retraining by enabling targeted forgetting and improved performance on the retain set, however the additional training may introduce computational overhead at scale.

Solutions that train with Differential Privacy (DP) (Dwork & Roth, 2014) can also be used for unlearning. DP-SGD (Abadi et al., 2016) masks the contribution of a single point by clipping gradients and injecting noise during training. As such, if an adversary cannot determine whether a point was used during training, unlearning it becomes unnecessary. Similar to exact unlearning methods, scaling DP-SGD can be challenging, while also maintaining high performance on the training data. Finally, Li et al. (2025) analyze *unnecessary unlearning* which also focuses on reducing the forget set prior to unlearning. However, our works maintain key differences in approach and empirical goals. First, their method relies on determining removal points using a cosine similarity feature matrix with similarity condition. In comparison, we use approximations of influence, which we find more effective than pure cosine similarity on our evaluation metrics (Appendix B and Figure B.3). Furthermore, our work shows the efficacy of forget set reductions for both vision and language domains, as well as, extends reductions to the retain set to accommodate unlearning algorithms that incorporate both the forget and retain sets.

### 2.0.2 Influence function approximations

Influence functions have a long history of use in statistics (Hampel, 1974). In practice, evaluating the classical influence function may be computationally infeasible which motivates the use of approximation methods. For example, Koh & Liang (2020) use second order optimization to approximate influence, and thus make it computationally cheaper, while retaining accuracy. In the setting of *targeted instructing tuning*, LESS (Xia et al., 2024b) uses random projections with LoRA to create a *gradient datastore*, a low dimensional representation of the gradient, and as such makes influence estimation tractable by using similarity. Feldman & Zhang (2020) provide a method that incorporates the influence of sets of points, while also providing a statistical guarantee that with high probability, the expectation of the estimated influence will differ from the ground truth by a small amount. TracIn (Pruthi et al., 2020) computes influence by tracing changes in loss to test points over training. Grosse et al. (2023) focus on scaling influence functions to LLMs by using Eigenvalue-corrected Kronecker-Factored Approximate Curvature (EK-FAC) approximation.

## 3 Methodology: using influence to find low-impact data

### 3.1 Hypothesis about Training Data

The goal of unlearning is to remove a forget set $S \subset D$ from a trained model. SOTA methods typically assume that all points in $S$ are equally important to learning. However, we posit the following: what if some points in $S$ did not meaningfully contribute to model training? We hypothesize that points in $S$ vary in model impact, and that there exists a subset of low-impact learning points in the forget set $S_{LI} \subset S$. To find these points, we propose using influence scores, which inherently measure impact on learning. In practice however, computing influence exactly is computationally infeasible (Equations 1.1-1.2). As such, we first

theoretically motivate our use of approximate influence functions by showing that they are tractable, as well as, have theoretical guarantees for small error under specific conditions.

## 3.2 Theoretical Motivation for Approximate Influence Functions

First, we develop a theoretical motivation for our use of approximate influence functions. Our argument is adapted from Broderick et al. (2023) who use approximate influence functions in order to detect outliers and to evaluate the sensitivity of statistical estimates to those outliers. The key idea is that approximate influence functions can be used to construct a first-order approximation of the test loss if specific points were removed. This approach provides a cost-efficient alternative to full retraining, enabling us to estimate the influence of individual examples—or sets of examples—on model performance and thereby assess whether they need to be unlearned.

First, let us introduce some notation. We have access to an observation $z_i$ which contains features and a label for each individual $i$. Observations make up training samples from $S_{train}$ and test samples from $S_{test}$. These sets are of size $n_{train}$ and $n_{test}$ respectively. We consider a forget set $\mathcal{S} \subset \mathcal{S}_{train}$. Let us generalize the problem of evaluating Equations 1.1 and 1.2 by considering a loss function $\ell$, so that $\ell(w; z_i)$ is the loss for individual $i$ if the model weights are set to $w \in \mathbb{R}^K$. Let $w^*$ be the weights we obtain if we train on the full training dataset and $w^*_{-\mathcal{S}}$ be the weights if we train only on those data points in $S_{train} \setminus \mathcal{S}$, i.e., the training data that is not in the forget set. Our goal is then to approximately evaluate the difference in average test loss after forgetting points $\mathcal{S}$,

$$\frac{1}{n_{test}} \sum_{i \in S_{test}} \ell(w^*; z_i) - \frac{1}{n_{test}} \sum_{i \in S_{test}} \ell(w^*_{-\mathcal{S}}; z_i). \tag{3.1}$$

As stated above, direct evaluation of (3.1) is not computationally feasible. In order to derive a more tractable approximation we first suppose that $w^*$ uniquely minimizes the training loss and that $w^*_{-\mathcal{S}}$ minimizes the training loss without points in the forget set. Formally,

$$w^* = \arg\min_{w \in \mathbb{R}^K} \sum_{i \in S_{train}} \ell(w; z_i), \quad w^*_{-\mathcal{S}} = \arg\min_{w \in \mathbb{R}^K} \sum_{i \in S_{train} \setminus \mathcal{S}} \ell(w; z_i).$$

In practice, training methods may not minimize the loss exactly, nor is there necessarily a unique global minimizer of the loss. Nonetheless, the above provides a useful approximation.

Now, following (Broderick et al., 2023), consider a set of weights $\alpha = \{\alpha_i\}_{i \in \mathcal{S}_{train}}$. Define a corresponding set of weights $w^*(\alpha)$ that minimize the weighted average training loss. That is

$$w^*(\alpha) = \arg\min_{w \in \mathbb{R}^K} \frac{1}{n_{train}} \sum_{i \in S_{train}} \alpha_i \ell(w; z_i),$$

where we assume that there exists a unique minimizer. Note then that $w^*$ is equal to $w^*(\alpha)$ in the special case in which $\alpha_i = 1$ for all $i$, which we write in shorthand as $\alpha = 1$. Moreover, letting $\alpha_{-\mathcal{S}}$ denote the $\alpha$ with $\alpha_i = 0$ for all $i$ in the forget set $\mathcal{S}$, and $\alpha_i = 1$ otherwise, we have $w^*_{-\mathcal{S}} = w^*(\alpha_{-\mathcal{S}})$.

We now apply the key step in our argument. Assuming both the loss function and $w^*(\alpha)$ are twice differentiable, we can apply a first-order Taylor approximation to get

$$\frac{1}{n_{test}} \sum_{i \in S_{test}} \ell(w^*_{-\mathcal{S}}; z_i) \approx \frac{1}{n_{test}} \sum_{i \in S_{test}} \ell(w^*; z_i) - \sum_{j \in \mathcal{S}} \frac{d}{d\alpha_j} \left( \frac{1}{n_{test}} \sum_{i \in S_{test}} \ell(w^*(\alpha); z_i) \right) \Big|_{\alpha=1},$$

and rearranging we then obtain

$$\frac{1}{n_{test}} \sum_{i \in S_{test}} \ell(w^*; z_i) - \frac{1}{n_{test}} \sum_{i \in S_{test}} \ell(w^*_{-\mathcal{S}}; z_i) \approx \sum_{j \in \mathcal{S}} \frac{d}{d\alpha_j} \left( \frac{1}{n_{test}} \sum_{i \in S_{test}} \ell(w^*(\alpha); z_i) \right) \Big|_{\alpha=1}.$$

That is, the difference in test loss with and without forgetting observations in $\mathcal{S}$ can be approximated by the sum of derivatives on the RHS above. The derivative with respect to $\alpha_i$ is the approximate influence function for individual $i$. It captures the impact on the optimized loss of a small change in the weight $\alpha_i$ placed on that individual's contribution to the training loss. As we show below, the approximate influence function can be computed straight-forwardly without any need for re-training.

Theorem 3.1 demonstrates that the approximate influence function has a tractable form that can be computed without re-training.The formula is given in terms of the Jacobians of the training and test losses, and the Hessian of the training loss. The Jacobians are defined by $J_j = \frac{\partial}{\partial w}\ell(w^*; z_j)$ and $\tilde{J} = \frac{1}{n_{test}}\sum_{i \in S_{test}} \frac{\partial}{\partial w}\ell(w^*; z_i)$ (i.e., $J$ is the length-$K$ vector whose $l$-th entry is $\frac{\partial}{\partial w_l}\ell(w^*; z_j)$). The Hessian $H$ is the $K$-by-$K$ matrix whose $(k, l)$-th entry is $\frac{1}{n_{train}}\sum_{i \in S_{train}} \frac{\partial^2}{\partial w_l w_k}\ell(w^*; z_i)$.

**Theorem 3.1.** *Suppose that for $\alpha$ sufficiently close to 1, that the function $w \mapsto \sum_{i \in S_{train}} \alpha_i \ell(w; z_i)$ is strictly convex and twice differentiable. It follows that,*

$$\frac{d}{d\alpha_j}\left(\frac{1}{n_{test}}\sum_{i \in S_{test}}\ell(w^*(\alpha); z_i)\right)\Bigg|_{\alpha=1} = -J_j' H^{-1} \tilde{J}.$$

The formula in Theorem 3.1 is a slight variation on the standard formula for the approximate influence function in M-estimation problems. Nonetheless, for completeness we provide a proof in Appendix C. The main computational challenge in the calculation of the approximate influence function is the Hessian $H$. However, this needs to be calculated only once, and only after initial training is complete. Overall, Theorem 3.1 shows that the influence function approximation has a tractable form.

### 3.3 Theoretical Guarantees for Influence Approximation

Broderick et al. (2023) provide conditions under which the approximation error resulting from the first order Taylor expansion is small, and thus the error incurred from the use of the approximate (rather than exact) influence is small. In particular, the error from the approximation is small when the forget set $\mathcal{S}$ contains only a relatively small fraction of the training data.

The theoretical guarantees in Broderick et al. (2023) build on results in Giordano et al. (2020). Adapted to our context, their results show that, under certain conditions, the error from the Taylor approximation to the exact influence of the forget set $\mathcal{S}$ is of order $(|\mathcal{S}|/n_{train})^2$, where $|\mathcal{S}|$ is the number of observations in the forget set. Moreover, this holds uniformly over all sufficiently small forget sets. The actual quantity to be approximated (the exact influence of the forget set $\mathcal{S}$) is generally of order $|\mathcal{S}|/n_{train}$. Thus, when only a small proportion of the training examples are to be forgotten, the approximation error is much smaller than the size of the object to be approximated. Below we adapt their results to our setting. While the proof follows similar steps to Giordano et al. (2020) we include a stand-alone proof in Appendix C.

To state the assumptions under which the result holds, define weight-specific Jacobians and Hessians by $J_i(w) = \frac{\partial}{\partial w}\ell(w; z_i)$ and $H_i(w) := \frac{\partial^2}{\partial w \partial w'}\ell(w; z_i)$ respectively (note that we continue to use the notation $J_i := J_i(w^*)$). For a vector $v$, $\|v\|$ denotes the Euclidean norm of the vector, and for a matrix $M$, $\|M\|_{op}$ is the operator norm (i.e., $\|M\|_{op} := \sup_{v:\|v\|=1}\|Mv\|$). We make the following assumptions. Note these are similar to those in Broderick et al. (2023).

**Assumption 1.** Let $\mathcal{W}$ be a convex set of weight vectors that contains $w^*(\alpha)$ for all binary vectors $\alpha$ with sufficiently many entries equal to 1.

i. There is a constant $c_{inv} < \infty$ so that

$$\sup_{w \in \mathcal{W}}\|\left(\frac{1}{n_{train}}\sum_{i \in S_{train}} H_i(w)\right)^{-1}\|_{op} \leq c_{inv}.$$

ii. There are constants $c_H, c_J < \infty$ so that for all $i \in \mathcal{S}_{train}$, $\sup_{w \in \mathcal{W}}\|H_i(w)\|_{op} \leq c_H$ and $\|J_i\| \leq c_J$.

iii. There are constants $\ell < \infty$ and $\Delta > 0$ so that

$$\sup_{w:\, 0<\|w-w^*\|\leq\Delta} \frac{\frac{1}{n_{train}}\sum_{i\in S_{train}}\|H_i(w)-H_i(w^*)\|_{op}}{\|w-w^*\|}\leq\ell.$$

Assumption 1 places regularity conditions on the Jacobian and Hessian of the loss function. Assumption 1.i states that the average Hessian is non-singular, with operator norm bounded above uniformly over all weight vectors in the set $\mathcal{W}$. Assumption 1.ii bounds the Hessian and Jacobian the former uniformly over weights in $\mathcal{W}$. Assumption 1.iii places a Lipschitz continuity condition on the average Hessian.

**Theorem 3.2.** *Suppose the conditions of Theorem 1 hold and Assumption 1 holds. Then there is a constant $C<\infty$ so that for any forget set $\mathcal{S}$ with $|\mathcal{S}|/n_{train}$ sufficiently small,*

$$|\frac{1}{n_{test}}\sum_{i\in S_{test}}\ell(w^*(\alpha_{-\mathcal{S}});z_i)-\frac{1}{n_{test}}\sum_{i\in S_{test}}\ell(w^*;z_i)-\tilde{J}H^{-1}\frac{1}{n_{test}}\sum_{i\in\mathcal{S}}J_i|$$
$$\leq C\big(\frac{|\mathcal{S}|}{n_{train}}\big)^2.$$

The theorem shows that the error from approximating the exact influence by the approximate influence for a forget set $\mathcal{S}$, is bounded by $C(|\mathcal{S}|/n_{train})^2$, uniformly over all small enough forget sets. By contrast, the approximate influence of $\mathcal{S}$, which is given by $\tilde{J}H^{-1}\frac{1}{n_{test}}\sum_{i\in\mathcal{S}}J_i$, is a sum over $\mathcal{S}$ terms and is scaled by $1/n_{test}$. Thus it will typically scale at rate $|\mathcal{S}|/n_{train}$, which is much larger than the approximation error when $|\mathcal{S}|/n_{train}$ is small.

In summary, under certain conditions, the approximation error incurred from using approximate influence is negligible (in an asymptotic sense, when the size of the forget set is small compared to the size of the training data). As such, this provides some guidance on the appropriateness of approximate influence for selecting forget sets. In a given practical application, however, the accuracy of the approximation remains an empirical question. Overall, these theoretical guarantees motivate our experiments in the remainder of the paper, and in particular the use of approximate rather than exact influence.

### 3.4 Empirical Evaluation

Given our theoretical motivation, we next explore several approximation methods in our experiments, and compare their efficacy. Specifically, we evaluate: (1) the Hessian approximation (Koh & Liang, 2017), (2) the LESS method (Xia et al., 2024a), and (3) Lowest Gradients, a heuristic based on low gradient norms. For our purposes, this last method is where we measure the extent to which the predicted soft logits on input $x$ are unchanged. The Hessian approximation most closely resembles our theoretically motivated approximation due to its use of Hessians and is popularly used alongside LESS. In contrast, Lowest Gradients is computationally cheaper. In turn, we provide a comparison across potential methods to weight potential advantages.

Koh & Liang show how changes in model predictions can be approximated using closed-form influence functions ($\mathcal{I}_{up,loss}(z,z_{test})$ involving the Hessian. To efficiently compute this influence, they use implicit Hessian-vector products (HVPs) to approximate the inverse of the Hessian, They also pre-compute these HVPs for each test point, and then reuse them for each training point. As a second order optimization problem, this approach approximates the exact definition of influence well.

Xia et al., focuses on adapting influence for selecting instruction fine-tuning data. LESS uses Adam instead of SGD, it performs data selection across sequences, and it uses a *gradient datastore* that scales well with large model size. The LESS method is useful for estimating influence in the setting of Large Language Models (LLMs), however the combination of random projections with Adam optimization makes it an efficient approximation for any prediction task. Unlike the Hessian estimation, the LESS method does not explicitly estimate leave-one-out influence as defined in Section 1.

In addition to other published methods of estimating influence, we also identify points in the training set where the $L2$ norm of the training gradient approaches 0 early on in training, and remains low throughout.

Methods such as the Hessian approximation use the training gradient as a target in influence estimation, thus it intuitively follows that points with small gradients throughout training would also have low training set influence. The principle advantage of using training gradients as proxies for influence is that they can be stored during initial model training and retrieved later on, making their use nearly free from a computational perspective. To better understand how early in training we should calculate the lowest gradients, we track their distribution throughout training (Appendix J).

### 3.4.1 Influence calculations and motivations

For the Hessian approximation and LESS, we compare two approaches for estimating point influence: (1) measuring the influence of training points on test set predictions (*test influence*), and (2) measuring the influence of training points on themselves (*self-influence*). The rationale for test influence is that a point with low influence on test predictions likely has limited impact on overall learning; thus, low-impact points $D_{LI}$ can be identified by computing their influence on the test set. For self-influence, the assumption is that a training point's greatest influence is on itself—if it has minimal self-influence, the model likely did not rely on it to generalize to that point, indicating low importance. Lastly, our Lowest Gradients heuristic builds on this intuition: if self-influence dominates, then low gradient magnitudes during training may correlate with low self-influence and thus indicate low-impact points.

### 3.4.2 Implementation

To compute influence using both test and training sets, as well as gradient information, we proceed as follows. First, we train a model on the full training set. For the Lowest Gradients heuristic, we record changes in input gradients during training, which serve as proxies for self-influence. The remaining methods are applied post-training. To estimate test influence, we compute the average influence of each training point on a representative subset of test examples. For self-influence, we measure the influence of each training point on itself. In all cases, the resulting influence scores are used to identify the low-impact subset $D_{LI}$. We also note additional details about our influence calculations in Appendix E.

## 4 Comparative analysis of influence approximation methods

In the following section, we empirically compare the ability of influence approximation methods to find low-impact impact training points. In particular, we investigate using the Hessian approximation (Koh & Liang, 2017) (*Hessian*), the LESS method (Xia et al., 2024a) (*LESS*), and the Lowest Gradients method (Section 3) (*Lowest Gradients*). For each method, we select a subset $D_{LI} \subset D$ of points using calculated influence scores, retrain a model on $D \setminus D_{LI}$, and then evaluate the retrained model's accuracy on $D_{LI}$ to measure deviations from the original model trained on $D$. Accuracy is a standard and easily interpretable metric that allows us to compare performance both across approximation methods, and tasks. We expect model accuracy on low-impact points $D_{LI}$ to be maintained regardless of whether they are included in training. Given this, such points are promising candidates for removal during unlearning, as their minimal contribution to model learning suggests they have little impact on unlearning (since they were never meaningfully learned)—a claim we later verify in Section 5.

### 4.0.1 Datasets

We incorporate both popular image and language datasets in our analysis. Specifically, we use **CIFAR-10** and **CIFAR-100**, that vary in class size and difficulty to test the generalizability of our method. The CIFAR-10 dataset Krizhevsky et al. (2009) is comprised of 60,000 training images distributed evenly over 10 classes. To test unlearning, a ResNet-101 He et al. (2016) model is trained for object detection using the provided training and testing sets. The features are computed using random cropping and horizontal image inversion, as well as standard transformer normalization. This model achieves a testing set prediction accuracy of 94% and a training set accuracy of 98%. For the Lowest Gradients method, we track the distribution of low gradient points (Appendix J) and use Checkpoint 5 in our results. CIFAR-100 Krizhevsky et al. (2009), on the other hand, has 100 classes with 20 superclasses. We again train a ResNet-101 model using the standard

train and test splits specified by the dataset. The training set accuracy for this model is 97% and a testing set accuracy of 92.7% We also use Checkpoint 5 for the Lowest Gradients method.

We additionally incorporate a popular language dataset to evaluate our methods in a different domain. In the **Stanford Question Answering Dataset (SQuAD)** Rajpurkar (2016), each example input consists of an article with $k$ sentences about a topic, and a question sentence. The output should be which sentence, if any, in the article answers the given question at any point in the sentence. There are several variant tasks for this dataset, such as the more difficult task of identifying the start and end tokens containing the answer. For the task model, we use a pre-trained BERT model to generate embeddings for both the target article sentences, and the question Devlin (2018), with an added ReLU activation layer. This task model achieves a training set accuracy of 0.83 and a testing accuracy of 0.77. For the Lowest Gradients approach, we also find Checkpoint 5 to be optimal.

## 4.1 Results

We first verify whether our influence approximation methods identify the same set of low influence points. Appendix Table 1 shows the Jaccard membership similarity and Spearman correlation coefficients of the various influence estimation methods when $|D_{LI}| = 14,000$, as well as similarity with data memorization scores[2]. We see that the LESS and Hessian methods show considerable overlap, particularly when self influence is used. The Lowest Gradients set also overlaps with these points substantially more than a randomly chosen collection.

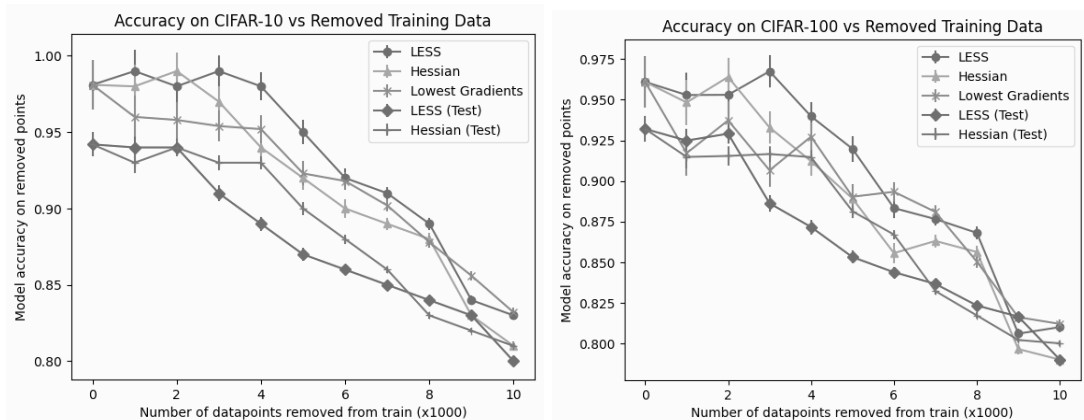

Figure 4.1: The model is retrained on all but the n lowest influence points (x-axis) from image datasets CIFAR-10 (**left**) and CIFAR-100 (**right**) and the final model accuracy on the removed points is recorded (y-axis). All influence methods are able to remove up to ∼2,000 points with minimal effects on accuracy, before this drops rapidly. The LESS and Hessian methods using self-influence consistently outperform the others.

Next we assess performance of each influence approximation method. Figures 4.1 and 4.2 show the retrained accuracy on a set $D_{LI}$ as a function of $|D_{LI}|$ for each influence scoring method. We note that the LESS method with self-influence generally performs best, being able to remove approximately 5,000 training points from CIFAR-10 and 8,000 points from SQuAD with negligible model degradation. This implies that LESS using self-influence may be an optimal choice for calculating influence across both language and vision. Additionally, the Lowest Gradients method of data selection outperforms LESS (test) and Hessian (test). This is impactful, because the Lowest Gradients method is computationally much cheaper to compute than the Hessian and LESS methods. As such, given resource-constraints, computing Lowest Gradients may be the optimal method for both language and vision domains. However, it should be considered that this is only the case when model training checkpoints are available and that it may not be possible to compute Lowest Gradients for pre-trained models. Finally, we analyze the distribution of low influence points by examining

---

[2]https://pluskid.github.io/influence-memorization/

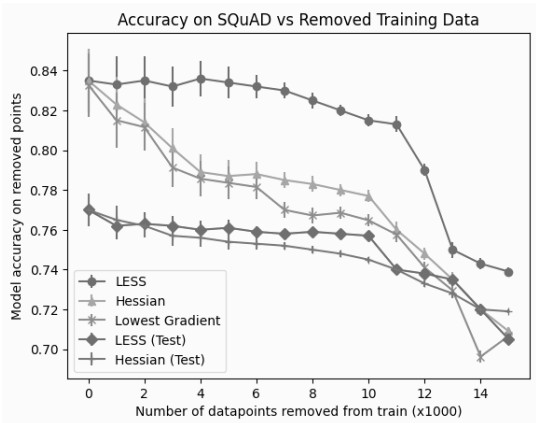

Figure 4.2: Using a setup similar to Figure 4.1, the model is now retrained on all but the n lowest influence points from language dataset SQuAD. Accuracy drops steadily for most methods up to ∼10,000 points removed. Furthermore, LESS and Hessian with self influence still outperform other methods.

class frequencies among the bottom 14,000 CIFAR-10 examples by influence method (Figure E.1). Both Hessian and LESS reveal similar patterns: low influence points are most common in the Plane, Car, Ship, and Truck classes—likely due to their distinct, easily learnable features. In contrast, animal classes like Cat, Deer, and Horse appear less frequently, suggesting their less easily distinguishable features contribute more significantly to model learning. In summary, we show that influence approximation methods can effectively identify subsets of low-impact points $D_{LI} \subset D$ such that models trained on $D \setminus D_{LI}$ generalize well to them. We also provide a comparison of these methods across tasks and set sizes to show performance variations.

## 4.2 Cost of Methods

In the previous section, we compare the performance of several influence approximation methods. Here, we analyze their associated costs, to better allow users to understand each method's trade-offs when integrating our unlearning framework (Section 5) with their specific use case. While exact runtime and memory requirements depend on hyperparameters (e.g. model size, dataset scale, etc...) we aim to contrast the relative overheads of the methods. **Lowest Gradients** incurs the least additional cost because it records gradient magnitudes during normal training. Since sample gradients may already be computed for parameter updates, the only extra requirement is storage and comparison of these values. The overhead is therefore minimal, although scaling datasets should be considered. However, this method relies on access to training checkpoints and thus may not be applicable for pre-trained models where such gradients are unavailable. In contrast, both **Hessian Approximation** and **LESS** require post-training computations. For Hessian Approximation, this involves using HVPs to estimate self-influence for each point, and can lead to a substantial computational cost through additional model passes. LESS, on the other hand, constructs a gradient datastore. As such, computing influence requires additional gradient and similarity computations, as well as, increased storage costs depending on the chosen projection dimension.

To provide additional guidance on the selection of an influence approximation method, we construct empirical upper and lower bounds on execution time based on our evaluations. In particular, we decompose the computational costs associated with calculating influence scores offline (prior to unlearning).

As a lower bound, we consider **Lowest Gradients**. As discussed previously, this method incurs negligible additional computational overhead because it records gradients during standard training. Consequently, the execution time cost of computing influence scores offline is effectively 0. Under this configuration, Lowest Gradients yields immediate execution time savings during unlearning. However, as stated previously, this method requires access to computed training gradients which may not be available.

As an upper bound, the **Hessian Approximation** computed using test points (rather than self-influence) is arguably our most computationally expensive method. This approach requires estimating the influence

of each training example on a subset of test points, resulting in substantially higher computational overhead. Using one H100 GPU (increasing this number can greatly reduce execution time), we measure the corresponding influence computation times (S) (see Table 11). Execution time is strongly dependent on dataset size, with our largest dataset, Yahoo Answers, requiring much more time than CIFAR-10. Likewise, potential data processing (i.e. image transformations) can also play a factor. Overall, these results highlight that the selection of which influence approximation to use is inherently user-centric. The appropriate influence approximation method should be selected based on expected performance requirements, and available computational resources, reflecting the trade-off between preprocessing cost and online unlearning efficiency.

## 5    Unlearning framework

Building on the previous section, where influence approximation methods can successfully identify low-impact training points, we now demonstrate how integrating these approximations into unlearning improves computational efficiency without compromising privacy or performance.

Recall that in a standard unlearning setup, the goal is to remove a forget set $S \subset D$ from a model trained on $D$, $\mathcal{A}(D)$, using an unlearning algorithm and potentially a retain set $R = D \setminus S$, $\mathcal{U}(\mathcal{A}(D), S, R)$. As $D$ scales, applying an unlearning algorithm to every point in $S$ becomes increasingly costly. This is particularly true for training points with minimal impact on model learning, $S_{LI} \subset S$, which may also have minimal effect on subsequent unlearning. Consequently, it is reasonable to exclude these points from unlearning if privacy and performance guarantees are preserved. Furthermore, many unlearning algorithms involve additional training on the retain set (Section 2.0.1). By the same logic, it is also reasonable to avoid unnecessary computations on low-impact retain set points, $R_{LI} \subset R$, when they do not meaningfully contribute to unlearning objectives.

Toward these points, we propose the following unlearning framework for influence function $\mathcal{I}$ and hyperparameter $x$:

1) Execute $\mathcal{I}(D, \mathcal{A}(D))$ and retrieve the influence of each point in $D$:  $d_i$

2) Sort $D$ by descending point influence $d_i$

3) Retrieve the bottom $x\%$ of lowest influence points in $D$:  $D_{LI}$

4) Remove low influence points from the forget and retain sets to create high influence subsets:
$$S_{HI} = S \setminus (S \wedge D_{LI})$$
$$R_{HI} = R \setminus (R \wedge D_{LI})$$

5) Apply $\mathcal{U}$ on $S_{HI}$ and $R_{HI}$:
$$\mathcal{U}(\mathcal{A}(D), S_{HI}, R_{HI})$$

Figure 5.1: Unlearning Framework.

We provide a guideline for choosing $x\%$ based on our empirical results in Appendix F.

### 5.1   Setup

We next evaluate our framework under three unlearning settings: (a) sample-wise unlearning, where either a specified forget set or a randomly selected subset of $D$ is unlearned, (b) class-wise unlearning, which requires forgetting all examples from a target class, and (c) subclass-wise unlearning, where targets from a specific subclass are unlearned. (Fan et al., 2025). We leverage the provided forget set and setup from the NeurIPS'23 competition on unlearning (see Section 4.1 for details) to test the performance of our unlearning framework in a sample-wise unlearning scenario. Then, to assess the generalizability of our framework, we extend our experiments to include evaluations on randomly selected subsets from CIFAR-100 (labels 0–49), Fashion-MNIST, and Yahoo Answers (data and training information in Appendix G). For class-wise unlearning, full classes from CIFAR-10 act as the forget sets. Finally, for subclass-wise unlearning, we unlearn full subclasses from CIFAR-100. Unlike CIFAR-10, CIFAR-100 contains more subclasses with fewer images

per class, allowing us to better test the robustness of our approach across datasets with greater variability. Fashion-MNIST, on the other hand, is larger than both CIFAR-10 and CIFAR-100 while still containing 10 classes (like CIFAR-10), allowing us to isolate the effect of scale. Finally, Yahoo Answers is a language dataset, enabling evaluation of our framework on language. It is also our largest dataset, with 1.4M training points, further testing the scalability of our method.

We begin by using the well-performing Hessian approximation to compute influence scores for all training points, using random subsets from the test set as evaluation points. In a setup similar to that of Section 4.1, we then demonstrate that retraining a model solely on high-influence points from CIFAR-10 preserves accuracy on the excluded low influence points (Figure I.1 (**top**)). While performance degrades gradually as more points are removed, it remains relatively stable compared to the *random* baseline—where the same number of points is removed at random—highlighting that low influence points contribute little to model performance.

We observe a similar trend when removing low influence points from the retain set (Figure I.1 (**bottom**)). In this case, performance degrades more sharply with increased point removal due to the larger absolute number of removed samples. Nonetheless, the model retains a consistent ability to correctly classify the removed low influence points, even when trained without them. These findings suggest that low influence points can be safely omitted with minimal impact on performance.

### 5.2 Results

We next integrate our unlearning framework with the top three ranked algorithms from the competition: *fanchuan* (Rank 1), *[kookmin Univ] LD&BGW&KJH* (Rank 2), and *Seif Eddine Achour* (Rank 3)[3]. These methods differ in how they utilize the forget and retain sets, providing a robust testbed for assessing our framework.

To ensure fair comparison in our recreated setup, we perform a hyperparameter sweep over learning rates and training epochs for each method (details in Appendix K). Furthermore, we also perform an ablation study to better understand the effect of certain hyperparameters on our algorithms (Appendix K.3), and describe our computing environment in Appendix L. Using the best-performing configurations, we compare trade-offs between performance and resource usage. Our results show that incorporating our unlearning strategy can significantly reduce computational costs (up to 50% of execution time) while maintaining competitive performance and privacy guarantees across our three unlearning scenarios. To measure these, we evaluate **accuracy** across evaluation datasets, resistance to a suite of **membership inference attacks (MIAs)**, as well as, performance on the competition's **official evaluation metric**[4].

#### 5.2.1 Accuracy

Using our described CIFAR-10 model, we apply each unlearning algorithm and measure execution time (in seconds) and final accuracy (computed as the average over runs (Appendix K)) on the competition's original forget and retain sets with progressively larger batches of these points removed (Figures 5.2, I.2 and I.3 and Tables 12, 15 and 16). In the baseline setup (*Original*), we apply unlearning to the full forget and retain sets, whereas *Bottom x%* refers to performance when an increasing proportion $x$ of low influence points are removed from the forget and/or retain set. Across algorithms, we observe that removing low influence points leads to **significant reductions** in execution time—up to 50% when points are removed from both sets—with **minimal drops** (given variation in runs) of final accuracy. The maintained accuracy on the forget (retain) set between our set-reduced models and the original unlearned model, also suggests preserved unlearning quality. Thus, beyond performance, this indicates that the original unlearning method's privacy guarantee carries over. We find consistent results when using randomly sampled forget sets from CIFAR-100 (Figure 5.3 and Table 13), Fashion-MNIST (Figure I.5 and Table 17), and Yahoo Answers (Figure I.6 and Table 18) and tracking performance changes. While the removal of low influence points from Fashion-MNIST has similar results to CIFAR-10, we find that less points can be removed from CIFAR-100 before inducing performance (and thus privacy) degradation. This is due to the class sizes of CIFAR-100 where each class

---

[3]https://www.kaggle.com/competitions/neurips-2023-machine-unlearning/leaderboard
[4]https://github.com/google-deepmind/unlearning_evaluation

is represented by significantly less points than in CIFAR-10, thereby making each individual image more impactful to learning. We verify this by tracking the distribution of influence scores for the bottom 5% of points in both CIFAR-10 and CIFAR-100 and confirm that CIFAR-100 has a more evenly spread distribution across larger influence values, demonstrating the larger impact each point has (Figure I.4).

We further extend our analysis to class-wise and subclass-wise unlearning. For CIFAR-10, we treat randomly selected classes as forget sets to unlearn (Table 19). We observe consistent behavior to before: accuracy on the forget set remains at 0% with minor variation to the test and retain accuracies, while execution time decreases substantially. Results for CIFAR-100 subclass-wise unlearning also show similar trends (Table 21). Overall, our results demonstrate that our unlearning framework provides a practical way to achieve substantial computational savings with minimal impact on model performance and degradation of unlearning quality through accuracy.

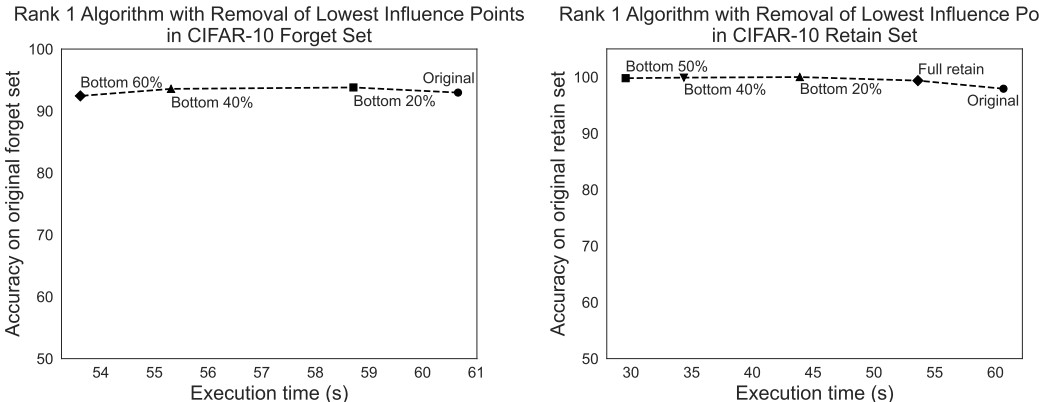

Figure 5.2: Proportions of low influence points are removed from the CIFAR-10 forget (**left**) and retain (**right**) sets using the influence scores on CIFAR-10 (sample-wise unlearning). These are then used in the Rank 1 unlearning algorithm. The final unlearned model's accuracy on the original set is recorded (y-axis) and compared to the total execution time (seconds) of executing unlearning (x-axis). Accuracy on the original sets remains about the same (with variations in runs), while execution time decreases as a larger proportion of points are removed before unlearning. The forget set accuracy does not drop to 0, unlike in the class-wise unlearning setting. In the sample-wise case, the forget set spans multiple classes and remains interspersed with highly similar retained examples, making complete forgetting substantially more difficult. Note that *Bottom 60%* in the left graph and *Full Retain* in the right graph are the same point as we continue to remove points from the retain set after removing from the forget set.

### 5.2.2 Membership inference attack (MIA)

We next further validate the claim that privacy is preserved when using our framework by analyzing changes in membership inference attack (MIA) accuracy. MIAs aim to determine whether specific data points were part of a model's training set, typically through an auxiliary model that detects differences in the target model's outputs (Shokri et al., 2017; Yeom et al., 2018; Salem et al., 2018).

Specifically, we use a family of six feature-based MIAs (*Loss MIA*, *Conf MIA*, *Entropy MIA*, *Margin MIA*, *TopK MIA*, *Combined MIA*). For a given unlearned model, a forget set, and a test set, we extract per-example features from the model outputs and evaluate whether these features allow an attacker to distinguish forget examples from test examples. Specifically, for each feature set, we train either a Random Forest or logistic-regression attack model, and report its membership-inference accuracy under cross-validation. (additional details on these features and the MIA implementations can be found in Appendix H). Our full MIA results for sample-wise unlearning using the CIFAR-10, CIFAR-100, Fashion-MNIST, and Yahoo Answers models/datasets can be found in Tables 2 to 8) respectively. Likewise, MIA results for class-wise unlearning on CIFAR-10 can be found in Table 20[5]

---

[5]Confidence intervals are computed using $1.96\times$ standard error.

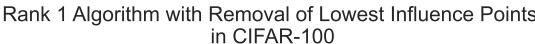
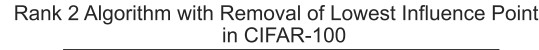
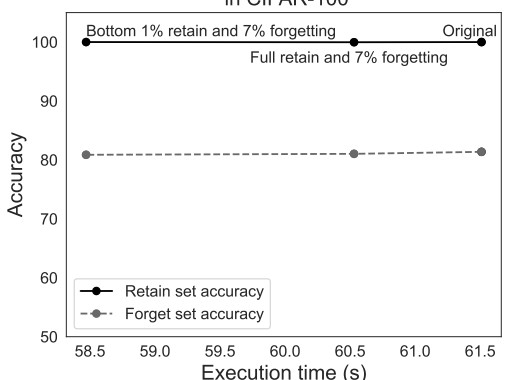
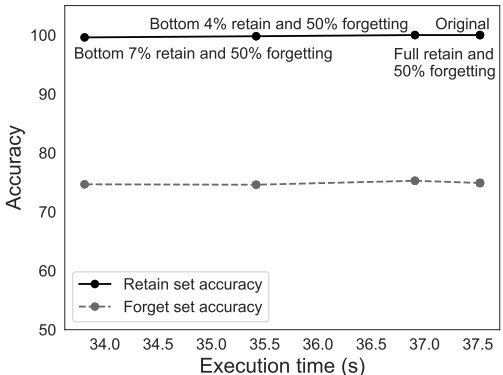

Figure 5.3: Proportions of low influence points are removed from the CIFAR-100 forget and retain sets before running the Rank 1 (**left**) and Rank 2 (**right**) algorithms (sample-wise unlearning). We compare only removing points from the forget set (*Full retain and x% forgetting*) to removing points from both sets simultaneously. We track performance on both the original forget and retain sets to ensure maintained performance in both. As can be seen, performance stays consistent.

Across a range of models and datasets, we consistently show that removing low influence points from the forget set—up to 60% of the lowest-ranked training points—reduces computational cost without degrading privacy. All of the MIA accuracies of the reduced unlearning models are within the standard error ranges of the original unlearned models, reinforcing our earlier findings: low influence points can be safely excluded from unlearning to decrease computational cost.

### 5.2.3 Competition Evaluation Metric

The NeurIPS'23 competition on unlearning introduces an evaluation framework that measures *forgetting quality* through a differential privacy (DP) inspired indistinguishability criterion, and then combines it with model utility and efficiency for a final score[6]. Specifically, an unlearning method $\mathcal{U}$ is evaluated by comparing the distribution of models produced by retraining on the retain set, $\mathcal{A}(D \setminus S)$, to the distribution produced by applying unlearning to the original model, $\mathcal{U}(\mathcal{A}(D), S, D)$. For each forget example $s$, scalar outputs $f(M(s))$ are collected from multiple retrained and unlearned models, and used to evaluate multiple decision rules that attempt to distinguish the two empirical output distributions. These estimated false positive and false negative rates, $\widehat{\text{FPR}}$ and $\widehat{\text{FNR}}$ are used to produce a privacy parameter $\hat{\varepsilon}$. The per-example value $\varepsilon_s$ is defined by the worst case attack. These per-example $\varepsilon_s$ values are then aggregated to produce the overall forgetting quality score $F$. Finally, $F$ is adjusted by the ratios of the retain and test accuracies of the unlearned models relative to the retrained models, while enforcing a hard efficiency cutoff on runtime to take into account model utility and efficiency.

In our experiments, we use the officially released implementation of this metric[7] as a base. However, instead of using CASIA-SURF(Zhang et al., 2020), which is inaccessible due to licensing restrictions, we implement two variations of the metric on CIFAR-10 and CIFAR-100 (Tables 9 and 10).These allow us to compute forgetting quality scores for unlearned models trained with and without low-influence points while preserving the competition metric's underlying logic. As can be seen, our baseline with access to the full forget set produces $F$ and final adjusted scores in the standard error ranges of unlearned models with influence-reduced forget sets. Furthermore, this result extends across both CIFAR-10 and CIFAR-100. To conclude, this metric does not show a discernible difference in forgetting quality when low influence points are removed using our framework.

---

[6]https://unlearning-challenge.github.io/assets/data/Machine_Unlearning_Metric.pdf
[7]https://github.com/google-deepmind/unlearning_evaluation

# 6 Conclusion

In this work, we provide a solution to reduce the computational costs of unlearning while maintaining privacy and performance. We first demonstrate that, across a range of models and tasks, there consistently exist subsets of training points—denoted $D_{LI} \subset D$—that have minimal impact on model performance. We explore influence-based techniques to efficiently identify these low-impact points and show that they can be safely removed without significantly affecting the model's generalization ability. Building on this insight, we propose an unlearning algorithm-agnostic framework that leverages influence approximations to reduce the sizes of both the forget and retain sets. Finally, we evaluate our framework in a real-world unlearning scenario, showing that it leads to substantial reductions in computational cost while maintaining the performance and privacy guarantees of the original unlearned model.

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

# A    Appendix

# B    Cosine similarity of nearest neighbors

## B.1    Description

We explore using cosine similarity of nearest neighbors to find minimal impact points by making the following assumption: training points with existing similar points in the training data, are more likely to have lower impact on learning, because the model is able to learn from their neighbors instead. As such, these points are less likely to offer unique knowledge, and thus impact learning significantly. Therefore, to find $S_{LI}$ in a forget set $S$, we can filter $S$ for similar points in $D \setminus S$, or in other words, similar points that would remain if the forget set $S$ were unlearned. By doing this, we guarantee that for each point in $S_{LI}$, there is a similar point in the training data that may allow the former to be less impactful.

To find similar points in the remaining training data, we use nearest neighbors with cosine similarity for every point in the forget set $S$ with $D \setminus S$. We then filter for $S_{LI}$ using a similarity threshold $c$, and by randomly sampling from those points. We also adjust this to incorporate the top $k$ nearest neighbors with cosine similarity $\geq c$.

## B.2    Results

In the following section, we set up experiments that incorporate nearest neighbors with cosine similarity to find $S_{LI}$. For our purposes, we set $S$ to be the whole training set. As such, we find the least impactful points in the training data. Then, we randomly sample and remove $n$ of these points using a similarity threshold $c$, where for each point removed, there are at least $k$ nearest neighbors with a similarity of $\geq c$ to that point remaining in the training data. We then train a new model on all but those points. Finally, we test the new model's ability to generalize on removed points.

We set up the described procedure for both an image classification task, Food-101 (Figure B.1) and a question and answering task, SQuAD (Figure B.2). Food-101 (Bossard et al., 2014) is a classification dataset featuring 101 food categories where each class offers 750 training and 250 reviewed test images. In Figure B.1, we fine-tune ViT pretrained on ImageNet-21k[8] on Food-101. We compare the test accuracy of the model on different removed sets with varying $c$ and $k = 1$ (*Nearest Neighbor Cosine Similarity*). There is a positive correlation with $c$ and the accuracy on the set removed, thereby indicating that points with more similar remaining nearest neighbors are better generalized to. As such, these points are less impactful on training, because the model is able to perform well on them without exposure. We compare our method to using a *Random* baseline where a group of the same number of points is removed randomly. Ultimately, our method does substantially better as $c$ increases. As such, using nearest neighbor with cosine similarity allows us to find $S_{LI}$, however increasing $c$ generally implies fewer available points. As such, we may not be able to remove as many points for consistently generalizable performance, which may be a significant constraint.

We find consistent results on language models when fine-tuning BERT Large on SQuAD (Figure B.2)(dataset description in Section 4). Using the same method of comparing test accuracy of the model on different removed sets with varying $c$ and $k = 1$, we see a similar positive correlation between $c$ and the accuracy on the set removed. Furthermore, this method allows for better generalization of the removed points compared to randomly removing points, as we demonstrate with the same *Random* baseline. Therefore, using nearest neighbors with cosine similarity is an effective method for finding groups of points that are less impactful on the learning of a model and as such, $S_{LI}$ for both vision and language tasks.

---

[8]https://huggingface.co/google/vit-base-patch16-224-in21k

Next, we compare using nearest neighbors with cosine similarity to approximations of influence for finding $S_{LI}$ (Figure B.3). We first calculate influence approximations (Koh & Liang, 2020) for a model trained on CIFAR-10 (dataset description in Section 4). We find that by computing average influence over a relatively small number of test points (e.g. 100), we are already able to identify low influence training points that can be removed and generalized on with 1.0 accuracy. This can save significantly on compute costs. Furthermore, when comparing our results to using nearest neighbors with cosine similarity and a random baseline, influence approximations outperforms both by identifying potential $S_{LI}$ that the model is able to more consistently generalize to. Therefore, using average influence over test points provides a more effective way to find $S_{LI}$.

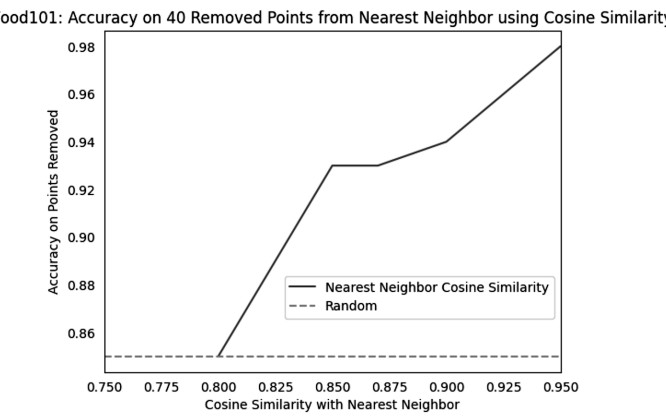

Figure B.1: We remove 40 random points from the Food-101 training set that have at least $k = 1$ neighbors with a cosine similarity of $\geq c$ (x-axis), and retrain a ViT model which we then test on the removed points (y-axis). We compare this to randomly removing 40 points with varying cosine similarity neighbors (*Random*) and testing accuracy on these points with a retrained model. As can be seen, the accuracy of the retrained model increases for removed points with higher cosine similarity with remaining neighbors.

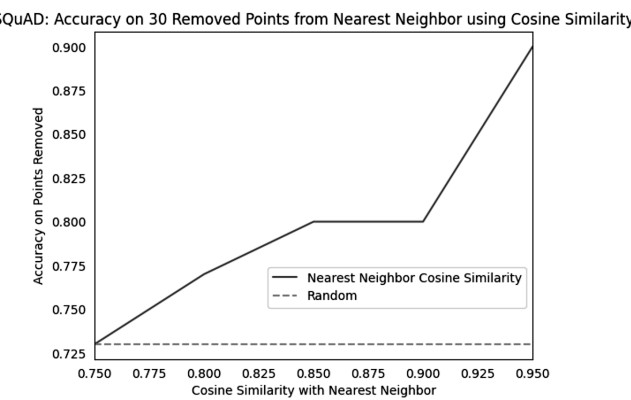

Figure B.2: Using a similar setup to Figure B.1, we measure the relationship between removing 30 points that have increasing cosine similarity with remaining training neighbors, and the effect this has on the retrained model's ability to answer the removed points correctly. However, in this graph we show that the positive relationship exists when retraining BERT Large on the language task, SQuAD.

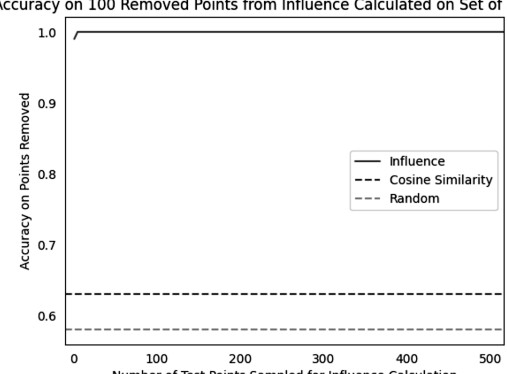

Figure B.3: We compare using influence approximations, cosine similarity, and random selection for finding low-impact points in CIFAR-10. Specifically, we use each of these methods to remove the same number of points (100) from a training set of 500 points, and then calculate a retrained model's accuracy on the points removed (y-axis). We show that typically influence outperforms using cosine similarity with near perfect accuracy on the removed points even with under 100 test points sampled for influence. Cosine similarity, however, still outperforms randomly selecting subsets to remove.

## C  Approximation influence proofs

*Proof of Theorem 3.1.* By the implicit function theorem,

$$\frac{d}{d\alpha_j}\left(\frac{1}{n_{test}}\sum_{i\in S_{test}}\ell\big(w^*(\alpha);z_i\big)\right)\bigg|_{\alpha=1}=\sum_{k=1}^{K}\frac{dw_k^*(\alpha)}{d\alpha_j}\bigg|_{\alpha=1}\tilde{J}, \tag{C.1}$$

where $w_k^*(\alpha)$ is the $k$-th component of the vector $w^*(\alpha)$.

By supposition, the minimization problem that defines $w^*(\alpha)$ is sufficiently regular that for any $\alpha$ in a neighborhood of 1, the following first-order conditions hold,

$$\frac{1}{n_{train}}\sum_{i\in S_{train}}\alpha_i\frac{\partial}{\partial w_l}\ell\big(w^*(\alpha);z_i\big)=0,\, l=1,2,...,K.$$

Differentiating both sides with respect to $\alpha_j$ gives for $l=1,2,...,K$,

$$\frac{d}{d\alpha_j}\left(\frac{1}{n_{train}}\sum_{i\in S_{train}}\alpha_i\frac{\partial}{\partial w_l}\ell\big(w^*(\alpha);z_i\big)\right)$$

$$=\frac{\partial}{\partial w_l}\ell\big(w^*(\alpha);z_j\big)+\sum_{k=1}^{K}\frac{dw_k^*(\alpha)}{d\alpha_j}\frac{1}{n_{train}}\sum_{i\in S_{train}}\alpha_i\frac{\partial^2}{\partial w_l w_k}\ell\big(w^*(\alpha);z_i\big)=0.$$

Evaluating the above at $\alpha=1$ we get $J_j+H\frac{dw^*(\alpha)}{d\alpha_j}\bigg|_{\alpha=1}=0$, where $\frac{dw^*(\alpha)}{d\alpha_j}$ is the length-$K$ vector whose $k$-th entry is $\frac{dw_k^*(\alpha)}{d\alpha_j}$. If $H$ is non-singular, then this implies $\frac{dw^*(\alpha)}{d\alpha_j}\bigg|_{\alpha=1}=-H^{-1}J_i$. Plugging this into (C.1) we get

$$\frac{d}{d\alpha_j}\left(\frac{1}{n_{test}}\sum_{i\in S_{test}}\ell\big(w^*(\alpha);z_i\big)\right)\bigg|_{\alpha=1}=-J_j'H^{-1}\tilde{J}.$$

$\square$

*Proof of Theorem 3.2.* Throughout we let $\alpha_i$ denote the $i$-th element of $\alpha_{-\mathcal{S}}$. By the first-order conditions and the remainder form of Taylor's theorem (which uses that $H_i$ is continuous), there is some $t \in [0,1]$ so that, letting $w_{t,\alpha} = tw^* + (1-t)w^*(\alpha_{-\mathcal{S}})$

$$
\begin{aligned}
0 &= \frac{1}{n_{test}} \sum_{i \in S_{test}} \alpha_i \frac{\partial}{\partial w} \ell\big(w^*(\alpha); z_i\big) \\
&= \frac{1}{n_{test}} \sum_{i \in S_{test}} \alpha_i J_i(w^*) \\
&\quad + \frac{1}{n_{test}} \sum_{i \in S_{test}} \alpha_i H_i(w_{t,\alpha}) dt \big(w^* - w^*(\alpha_{-\mathcal{S}})\big).
\end{aligned}
$$

Rearranging and adding and subtracting terms, and using the first order conditions $\frac{1}{n_{test}} \sum_{i \in S_{test}} J_i(w^*) = 0$, we get

$$
\begin{aligned}
&w^*(\alpha_{-\mathcal{S}}) - w^* \\
&= \left(\frac{1}{n_{test}} \sum_{i \in S_{test}} \alpha_i H_i(w_{t,\alpha})\right)^{-1} \frac{1}{n_{test}} \sum_{i \in S_{test}} \alpha_i J_i(w^*) \\
&= -\left[\left(\frac{1}{n_{test}} \sum_{i \in S_{test}} \alpha_i H_i(w_{t,\alpha})\right)^{-1} - \left(\frac{1}{n_{test}} \sum_{i \in S_{test}} H_i(w^*)\right)^{-1}\right] \frac{1}{n_{test}} \sum_{i \in S_{test}} (1 - \alpha_i) J_i(w^*) \\
&\quad - \left(\frac{1}{n_{test}} \sum_{i \in S_{test}} H_i(w^*)\right)^{-1} \frac{1}{n_{test}} \sum_{i \in S_{test}} (1 - \alpha_i) J_i(w^*).
\end{aligned}
\tag{C.2}
$$

Now, by the triangle inequality, the fact that $\alpha_i \in [0,1]$, and the definition of the operator norm, we have

$$
\begin{aligned}
&\|w^*(\alpha_{-\mathcal{S}}) - w^*\| \\
&\leq \frac{1}{n_{test}} \sum_{i \in S_{test}} (1 - \alpha_i) \|J_i(w^*)\| \\
&\quad \times \left[\left\|\left(\frac{1}{n_{test}} \sum_{i \in S_{test}} H_i(w_{t,\alpha})\right)^{-1}\right\|_{op} + \left\|\left(\frac{1}{n_{test}} \sum_{i \in S_{test}} H_i(w^*)\right)^{-1}\right\|_{op}\right. \\
&\quad \left. + \left\|\left(\frac{1}{n_{test}} \sum_{i \in S_{test}} H_i(w_{t,\alpha})\right)^{-1} - \left(\frac{1}{n_{test}} \sum_{i \in S_{test}} \alpha_i H_i(w_{t,\alpha})\right)^{-1}\right\|_{op}\right] \\
&\quad + \left\|\left(\frac{1}{n_{test}} \sum_{i \in S_{test}} H_i(w^*)\right)^{-1}\right\|_{op} \frac{1}{n_{test}} \sum_{i \in S_{test}} (1 - \alpha_i) \|J_i(w^*)\|.
\end{aligned}
$$

By Assumptions 1.i and ii. , and using $\alpha_i = \mathbb{1}\{i \in \mathcal{S}\}$, we get

$$
\begin{aligned}
&\|w^*(\alpha_{-\mathcal{S}}) - w^*\| \\
&\leq 3 \frac{c_{inv} c_J |\mathcal{S}|}{n_{test}} \\
&\quad + \frac{c_J |\mathcal{S}|}{n_{test}} \left\|\left(\frac{1}{n_{test}} \sum_{i \in S_{test}} H_i(w_{t,\alpha})\right)^{-1} - \left(\frac{1}{n_{test}} \sum_{i \in S_{test}} \alpha_i H_i(w_{t,\alpha})\right)^{-1}\right\|_{op}
\end{aligned}
$$

Consider the term in the final line. This term is of the form $\|A^{-1} - B^{-1}\|_{op}$ for two matrices $A$ and $B$. Note that for any two non-singular matrices $A$ and $B$, if $\|A - B\|_{op}$ is sufficiently small so that $\|A^{-1}\|_{op}\|A - B\|_{op} \leq$

$\frac{1}{2}$, then $\|A^{-1} - B^{-1}\|_{op} \leq 2\|A^{-1}\|_{op}^2\|A - B\|_{op}$.[9] In our case note that $\|A^{-1}\|_{op}^2 \leq c_{inv}^2$ by Assumption 1.i, and thus it suffices to derive an upper bound for $\|A - B\|_{op}$ and show this goes to zero as $\frac{|\mathcal{S}|}{n_{test}} \to 0$. Note that

$$\|\frac{1}{n_{test}} \sum_{i \in S_{test}} \alpha_i H_i(w_{t,\alpha}) - \frac{1}{n_{test}} \sum_{i \in S_{test}} H_i(w_{t,\alpha})\|_{op}$$

$$= \|\frac{1}{n_{test}} \sum_{i \in S_{test}} (\alpha_i - 1) H_i(w_{t,\alpha})\|_{op}$$

$$\leq \frac{1}{n_{test}} \sum_{i \in S_{test}} (\alpha_i - 1)\|H_i(w_{t,\alpha})\|_{op}$$

$$\leq \frac{|\mathcal{S}|}{n_{test}} c_H$$

So in all, if $\frac{|\mathcal{S}|}{n_{test}}$ is sufficiently small, then for some constant $C_1$,

$$\|w^*(\alpha_{-\mathcal{S}}) - w^*\| \leq 3c_{inv}c_J \frac{|\mathcal{S}|}{n_{test}} + 2c_{inv}^2 c_J c_H \left(\frac{|\mathcal{S}|}{n_{test}}\right)^2$$

$$\leq C_1 \frac{|\mathcal{S}|}{n_{test}}$$

Now, returning again to C.2 and again applying the triangle inequality and definition of the operator norm, we see that

$$\|w^*(\alpha_{-\mathcal{S}}) - w^* + \left(\frac{1}{n_{test}} \sum_{i \in S_{test}} H_i(w^*)\right)^{-1} \frac{1}{n_{test}} \sum_{i \in S_{test}} (1 - \alpha_i) J_i(w^*)\|$$

$$\leq \|\left(\frac{1}{n_{test}} \sum_{i \in S_{test}} \alpha_i H_i(w_{t,\alpha})\right)^{-1} - \left(\frac{1}{n_{test}} \sum_{i \in S_{test}} H_i(w^*)\right)^{-1}\| \frac{c_J |\mathcal{S}|}{n_{test}} \qquad \text{(C.3)}$$

Now, the term on the RHS above is again of the form $\|A^{-1} - B^{-1}\|$ and so we will again upper bound $\|A - B\|_{op}$ and show this goes to zero as $\frac{|\mathcal{S}|}{n_{test}} \to 0$. Using the triangle inequality and $\alpha_i \in [0, 1]$ we get

$$\|\frac{1}{n_{test}} \sum_{i \in S_{test}} \alpha_i H_i(w_{t,\alpha}) - \frac{1}{n_{test}} \sum_{i \in S_{test}} H_i(w^*)\|_{op}$$

$$\leq \frac{1}{n_{test}} \sum_{i \in S_{test}} \|H_i(w_{t,\alpha}) - H_i(w^*)\|_{op}$$

$$+ \frac{1}{n_{test}} \sum_{i \in S_{test}} (1 - \alpha_i)\|H_i(w^*)\|_{op}.$$

From $\|H_i(w^*)\|_{op} \leq c_H$ we have

$$\frac{1}{n_{test}} \sum_{i \in S_{test}} (1 - \alpha_i)\|H_i(w^*)\|_{op} \leq \frac{|\mathcal{S}|}{n_{train}} c_H,$$

and by Assumption 1.iii, we have

---

[9]To be precise, if $\|A^{-1}\|_{op}\|A - B\|_{op} < 1$ then $\|A^{-1} - B^{-1}\|_{op} \leq \frac{\|A^{-1}\|_{op}^2\|A - B\|_{op}}{1 - \|A^{-1}\|_{op}\|A - B\|_{op}}$.

$$\frac{1}{n_{test}} \sum_{i \in S_{test}} \|H_i\big(t^*w^* + (1-t^*)w^*(\alpha_{-\mathcal{S}})\big) - H_i(w^*)\|_{op}$$

$$\leq \frac{1}{n_{test}} \sum_{i \in S_{test}} (t^* - 1)\ell\|w^* - w^*(\alpha_{-\mathcal{S}})\|$$

$$\leq \ell\|w^* - w^*(\alpha_{-\mathcal{S}})\|.$$

Putting these facts together we arrive at

$$\|\frac{1}{n_{test}} \sum_{i \in S_{test}} \alpha_i H_i\big(t^*w^* + (1-t^*)w^*(\alpha_{-\mathcal{S}})\big) - \frac{1}{n_{test}} \sum_{i \in S_{test}} H_i(w^*)\|_{op}$$

$$\leq \ell\|w^* - w^*(\alpha_{-\mathcal{S}})\| + \frac{|\mathcal{S}|}{n_{train}} c_H$$

$$\leq (\ell C_1 + c_H)\frac{|\mathcal{S}|}{n_{test}}$$

where the final inequality used our our earlier upper bound on $\|w^* - w^*(\alpha_{-\mathcal{S}})\|$. So we see there is a constant $C_2 < \infty$ so that if $\frac{|\mathcal{S}|}{n_{train}}$ is sufficiently small, then using our earlier notation,

$$\|w^*(\alpha_{-\mathcal{S}}) - w^* - H^{-1}\frac{1}{n_{test}} \sum_{i \in \mathcal{S}} J_i(w^*)\|$$

$$\leq c_J c_{inv}^2 (\ell C_1 + c_H)\big(\frac{|\mathcal{S}|}{n_{test}}\big)^2$$

$$= C_2\big(\frac{|\mathcal{S}|}{n_{test}}\big)^2.$$

Now again by Taylor's remainder theorem, for some $\tilde{t} \in [0,1]$, letting $\tilde{w}_{t,\alpha} = \tilde{t}w^* + (1-\tilde{t})w^*(\alpha_{-\mathcal{S}})$ we have

$$\frac{1}{n_{test}} \sum_{i \in S_{test}} \ell(w^*(\alpha_{-\mathcal{S}}); z_i) - \frac{1}{n_{test}} \sum_{i \in S_{test}} \ell(w^*; z_i)$$

$$= \frac{1}{n_{test}} \sum_{i \in S_{test}} J_i(\tilde{w}_{t,\alpha})\big(w^*(\alpha_{-\mathcal{S}}) - w^*\big)$$

$$= \frac{1}{n_{test}} \sum_{i \in S_{test}} \Big(J_i(\tilde{w}_{t,\alpha}) - J_i(w^*)\Big)\big(w^*(\alpha_{-\mathcal{S}}) - w^*\big)$$

$$+ \tilde{J}\big(w^*(\alpha_{-\mathcal{S}}) - w^* - H^{-1}\frac{1}{n_{test}} \sum_{i \in \mathcal{S}} J_i(w^*)\big)$$

$$+ \tilde{J}H^{-1}\frac{1}{n_{test}} \sum_{i \in \mathcal{S}} J_i(w^*).$$

Using the triangle inequality and definition of the operator norm, we then get,

$$\|\frac{1}{n_{test}}\sum_{i\in S_{test}}\ell(w^*(\alpha_{-\mathcal{S}});z_i) - \frac{1}{n_{test}}\sum_{i\in S_{test}}\ell(w^*;z_i) - \tilde{J}H^{-1}\frac{1}{n_{test}}\sum_{i\in \mathcal{S}}J_i(w^*)\|$$

$$\leq \frac{1}{n_{test}}\sum_{i\in S_{test}}\|J_i(\tilde{w}_{t,\alpha}) - J_i(w^*)\|\|w^*(\alpha_{-\mathcal{S}}) - w^*\|$$

$$+\|\tilde{J}\|\|w^*(\alpha_{-\mathcal{S}}) - w^* - H^{-1}\frac{1}{n_{test}}\sum_{i\in \mathcal{S}}J_i(w^*)\|$$

$$\leq C_1\frac{|\mathcal{S}|}{n_{test}}\Big(\frac{1}{n_{test}}\sum_{i\in S_{test}}\|J_i(\tilde{w}_{t,\alpha}) - J_i(w^*)\|\Big)$$

$$c_J C_2\big(\frac{|\mathcal{S}|}{n_{test}}\big)^2$$

Where the final inequality uses our earlier results and Assumption 1.ii. Finally, by Assumption 1.ii it follows that

$$\|J_i(\tilde{w}_{t,\alpha}) - J_i(w^*)\| \leq c_H\|\tilde{w}_{t,\alpha} - w^*\|$$
$$= c_H\|w^* - w^*(\alpha_{-\mathcal{S}})\|$$
$$\leq c_H C_1\frac{|\mathcal{S}|}{n_{test}}$$

And so in all, for some constant $C$,

$$\|\frac{1}{n_{test}}\sum_{i\in S_{test}}\ell(w^*(\alpha_{-\mathcal{S}});z_i) - \frac{1}{n_{test}}\sum_{i\in S_{test}}\ell(w^*;z_i) - \tilde{J}H^{-1}\frac{1}{n_{test}}\sum_{i\in \mathcal{S}}J_i(w^*)\|$$

$$\leq (c_H C_1^2 + c_J C_2)\big(\frac{|\mathcal{S}|}{n_{test}}\big)^2$$

$$\leq C\big(\frac{|\mathcal{S}|}{n_{test}}\big)^2$$

$\square$

## D   Influence unlearning effects

---
**Algorithm 1** Computing accuracy cost of unlearning with influence
---
$D \leftarrow$ Full Dataset
$influence\_scores \leftarrow influence\_function(D)$
$low\_influence \leftarrow sort(D, influence\_scores)$
**while** $removed\_count \leq removed\_threshold$ **do**
    $D_{low} \leftarrow D - low\_influence[0, removed\_count]$
    $model \leftarrow train\_model(D_{low})$
    $accuracy \leftarrow append(accuracy, error(model, test\_set)$
    $removed\_count \leftarrow removed\_count + stepsize$
**end while**

---

The process described in Algorithm 1 generates a sequence of accuracy values that represents the testing set loss when removing low influence sets of datapoints. The function $influence\_function(D)$ takes a dataset $D$ and returns a real valued influence score for each sample, with smaller values indicating less influence, (e.g. Hessian estimates, LESS, or Lowest Gradients values).

Table 1: Jaccard set similarity of lowest 14k influence points in CIFAR-100 across influence approximation methods, and Spearman correlation coefficient in parentheses. The two Gradient methods are correlated, as expected. Interestingly, the Hessian method is also correlated with the Gradient methods.

| Set Similarity on CIFAR-100 | | | | | | |
|---|---|---|---|---|---|---|
| | **Hessian** | **LESS** | **Gradient (L2)** | **Gradient (L-Inf)** | **Memorization** | **Random** |
| **Hessian** | 1 (1) | 0.21 (0.24) | 0.37 (0.41) | **0.38 (0.39)** | 0.37 (0.35) | 0.04 (-0.03) |
| **LESS** | 0.21 (0.24) | 1 (1) | 0.21 (0.27) | 0.29 (0.31) | 0.26 (0.24) | 0.13 (0.08) |
| **Gradient (L2)** | 0.37 (0.41) | 0.21 (0.27) | 1 (1) | **0.89 (0.85)** | 0.31 (0.35) | 0.11 (0.09) |
| **Gradient (L-Inf)** | **0.38 (0.39)** | 0.29 (0.31) | **0.89 (0.85)** | 1 (1) | 0.33 (0.28) | 0.10 (0.13) |
| **Memorization** | 0.37 (0.35) | 0.26 (0.24) | 0.31 (0.35) | 0.33 (0.28) | 1 (1) | 0.09 (0.05) |
| **Random** | 0.04 (-0.03) | 0.13 (0.08) | 0.11 (0.09) | 0.10 (0.13) | 0.09 (0.05) | 1 (1) |

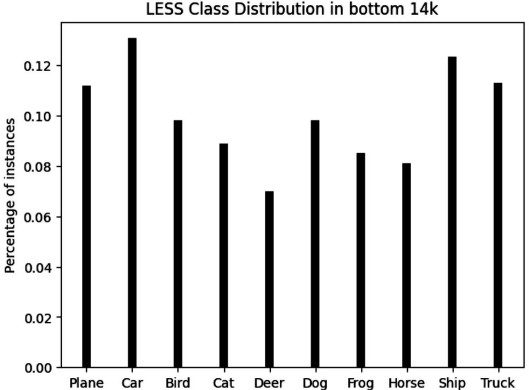 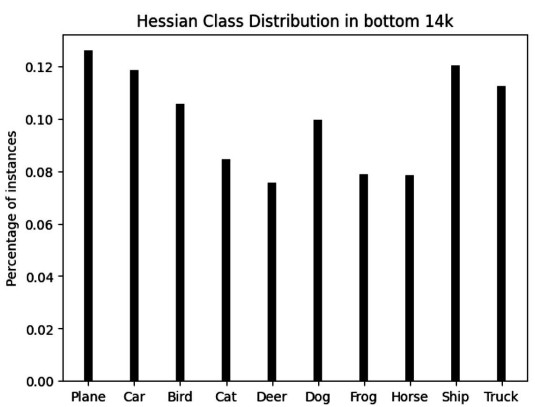

Figure E.1: We categorize the distribution of the bottom 14,000 of low influence points in CIFAR-10 by their classes. Both influence functions LESS self-influence (**left**) and Hessian self-influence (**right**) tend to have similar class distributions with low influence. Specifically, both have a large number of transportation vehicles like Plane, Car, Ship and Truck. Likewise, animals like Cat, Deer and Horse are less likely to have low influence.

# E  Notes on influence calculations

We also make the following notes about our methodology:

- While the definition of influence applies to the removal of a single point, low individual influence does not guarantee that a group of such points will have low collective influence. For example, duplicated points may each have low individual influence, yet their combined removal could significantly impact learning. In this work, however, we focus on identifying and analyzing points that exert influence independently, where the impact of removal is primarily driven by the point itself.

- We acknowledge that the order in which training points are presented can affect their measured influence—for example, among two similar data points, the one seen earlier during training may exert greater influence. However, in realistic scenarios where the training order is fixed and only the final model is accessible, this limitation is often unavoidable. Accordingly, our approach estimates influence relative to a specific training regime, and we do not claim that the resulting influence rankings would remain consistent under different hyperparameters or training conditions.

## F   Choosing $x$

While choosing $x\%$ for retrieving the lowest-influence points from $D$ depends on setup-specific factors, our empirical results can be used to provide a basic guideline. Across both vision and language datasets of varying sizes (Section 4, Section 5), we observe that selecting $x \leq 4$ is generally safe, and this holds consistently across influence approximation methods. We therefore recommend $x = 4$ as a starting point. Depending on dataset scale and resource constraints, this value can then be increased in increments of 5–10, monitoring degradation in unlearning metrics (potentially through approximations to save further on cost) to balance execution cost savings with performance preservation.

## G   Additional datasets and models

**Fashion-MNIST(Xiao et al., 2017):** an image classification dataset consisting of 60,000 training examples and 10,000 test examples, where each image is labelled with one of 10 clothing-related classes. For our framework experiments, we initialize from a publicly available pretrained model, `kkkkkgsp/my_awesome_fashion_model`[10], which is a fine-tuned version of `google/vit-base-patch16-224-in21k` on Fashion-MNIST. To obtain stronger baseline performance prior to unlearning, we further fine-tune this model for an additional 5 epochs on the Fashion-MNIST training set.

**Yahoo Answers(Zhang et al., 2016):** a large-scale text classification benchmark comprising 1.4 million training examples and 60,000 test examples. Each example contains a question title and question content, and the task is to predict one of 10 topic labels (indexed 0–9). We use `distilbert/distilbert-base-uncased`[11] as our base language model and fine-tune it for 3 epochs on the Yahoo Answers training set.

## H   MIA descriptions

*Loss MIA* uses the per-sample cross-entropy loss as a 1D feature and checks whether forget and test points remain separable by their loss values from the unlearned model. *Confidence MIA* uses the maximum softmax probability computed from logits and checks whether membership can be inferred from differences in the model's peak predicted probability. *Entropy MIA* uses the predictive entropy of the softmax distribution to test whether the overall uncertainty of the model's predictive distribution differs between forget and test points. *Margin MIA* uses the top-1 − top-2 probability gap and checks for separability based on how sharply the model prefers the top-1 vs the top-2 class. *Top-k MIA* uses the vector of the top-k softmax probabilities as a multi-dimensional feature and tests whether the shape of the model's highest-probability mass differs between forget and test examples. Finally, *Combined MIA* concatenates multiple extracted quantities from the previously described MIAs into a single feature vector, and checks whether a classifier can exploit any joint signal across these statistics to distinguish forget from test samples.

---

[10]https://huggingface.co/kkkkkgsp/my_awesome_fashion_model
[11]https://huggingface.co/distilbert/distilbert-base-uncased

# I  Unlearning framework: additional results

Table 2: Loss MIA accuracy (logistic-regression) on the unlearned model after applying Rank 1 and Rank 2 algorithms respectively. As more low influence forget set points are removed, MIA accuracy remains within the standard error range. This implies that the removal of low influence points reduces computational expenses, while maintaining the privacy guarantees of the original unlearned model. Confidence intervals are computed using $1.96\times$ standard error.

| Loss MIA Accuracy on CIFAR-10 | | |
|---|---|---|
| | **Rank 1** | **Rank 2** |
| Original | $.509 \pm .009798$ | $.522 \pm .009790$ |
| Bottom 20% | $.505 \pm .009800$ | $.518 \pm .009793$ |
| Bottom 40% | $.508 \pm .009799$ | $.514 \pm .009796$ |
| Bottom 60% | $.507 \pm .009799$ | $.519 \pm .009793$ |

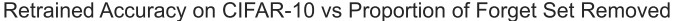

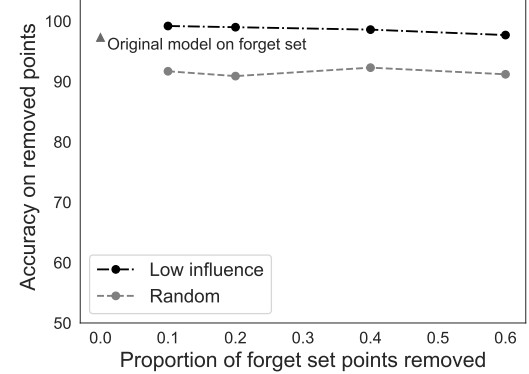

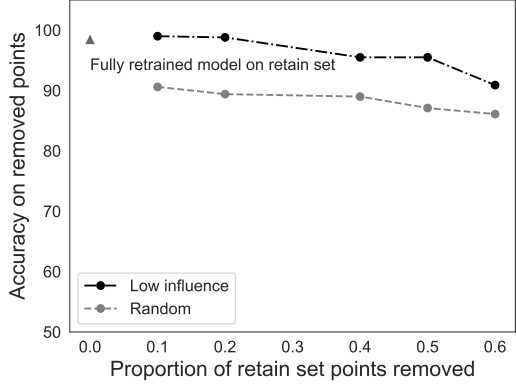

Figure I.1: Similar to Figure 4.1, the model is retrained on all but an n-proportion of the CIFAR-10 forget (**top**) or retain (**bottom**) set (x-axis). Accuracy on the removed points is then recorded after training (y-axis). In both graphs, a significant portion of points can be removed while maintaining retrained model performance. In contrast, a baseline of removing n-proportion of the forget (retain) set randomly performs strictly worse.

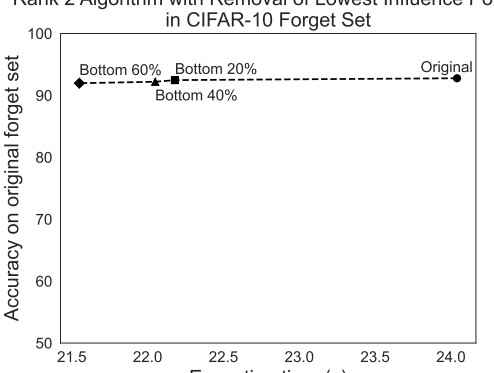 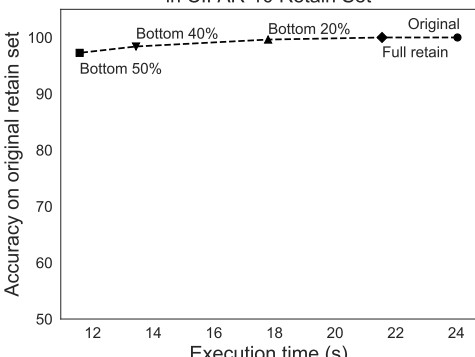

Figure I.2: Similarly to Figure 5.2, we remove points from the forget (**left**) and retain (**right**) sets using the same low influence proportions and track changes in performance on the original set, as well as execution time when using the Rank 2 algorithm (sample-wise unlearning). A similar pattern emerges where execution time of unlearning rapidly decreases with minimal changes to set accuracies from the original unlearned model. Note that *Bottom 60%* in the left graph and *Full Retain* in the right graph are the same point as we continue to remove points from the retain set after removing from the forget set.

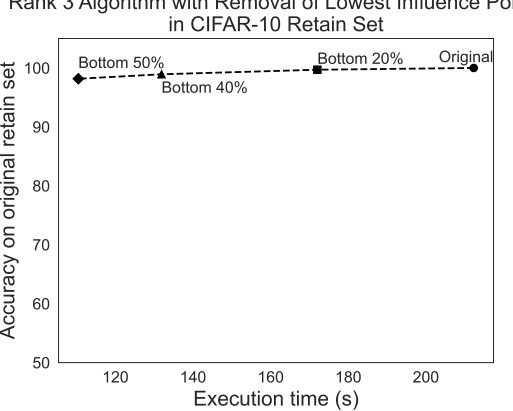

Figure I.3: Similarly to Figure 5.2, we remove points from the retain set using the same low influence proportions and track changes in performance on the retain set, as well as execution time on the Rank 3 algorithm (sample-wise unlearning). A similar pattern emerges where execution time of unlearning rapidly decreases with minimal changes to retain set accuracy from the original unlearned model.

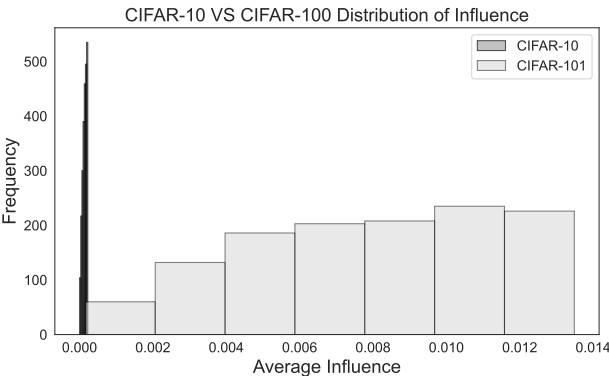

Figure I.4: We track the distribution of average influence for the bottom 5% of training points across both CIFAR-10 and CIFAR-100. While the bottom points of CIFAR-10 are concentrated around 0, indicating negligible influence, the distribution of CIFAR-100 is more evenly distributed and covers a greater range of larger influence values.

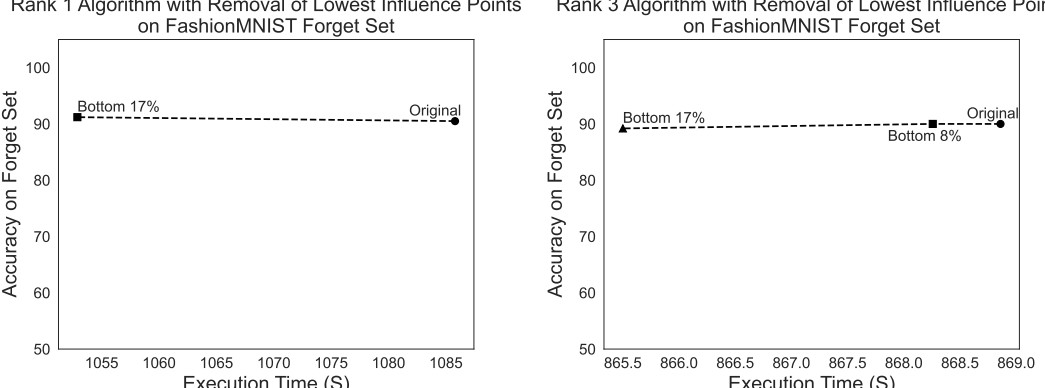

Figure I.5: Similarly to Figure 5.2, we remove points from the forget sets of Fashion-MNIST and perform unlearning using the Rank 1 (**left**) and Rank 3 (**right**) algorithms, and track changes in execution time (sample-wise unlearning). As previously, execution time decreases with minimal changes to set accuracies from the original unlearned model.

Table 3: **Additional MIA tests for CIFAR-10 (logistic-regression):** Conf MIA, Entropy MIA, Margin MIA, TopK (3) MIA, and Combined MIA values for the unlearned models when they unlearn on the full forget sets, compared to increasingly reduced forget sets for both the Rank 1 and Rank 2 unlearning algorithms (sample-wise unlearning). For both algorithms, each of the full forget and decreased forget set MIA standard error ranges overlap, indicating no discernible difference.

| MIA Type | Rank 1 | | | Rank 2 | | |
|---|---|---|---|---|---|---|
| | **Full** | **Bottom 20%** | **Bottom 50%** | **Full** | **Bottom 20%** | **Bottom 50%** |
| Conf MIA | $0.509 \pm 0.011$ | $0.502 \pm 0.006$ | $0.498 \pm 0.006$ | $0.503 \pm 0.008$ | $0.517 \pm 0.006$ | $0.512 \pm 0.009$ |
| Entropy MIA | $0.501 \pm 0.011$ | $0.487 \pm 0.008$ | $0.495 \pm 0.010$ | $0.510 \pm 0.010$ | $0.511 \pm 0.011$ | $0.512 \pm 0.008$ |
| Margin MIA | $0.508 \pm 0.009$ | $0.497 \pm 0.007$ | $0.494 \pm 0.007$ | $0.508 \pm 0.005$ | $0.512 \pm 0.006$ | $0.516 \pm 0.008$ |
| TopK(3) MIA | $0.512 \pm 0.007$ | $0.503 \pm 0.006$ | $0.506 \pm 0.009$ | $0.511 \pm 0.005$ | $0.509 \pm 0.005$ | $0.510 \pm 0.007$ |
| Combined MIA | $0.459 \pm 0.012$ | $0.476 \pm 0.013$ | $0.463 \pm 0.009$ | $0.485 \pm 0.003$ | $0.494 \pm 0.005$ | $0.481 \pm 0.005$ |

Table 4: **MIA tests for CIFAR-10 (Random Forest):** Loss MIA, Conf MIA, Entropy MIA, Margin MIA, TopK (3) MIA, and Combined MIA values for the CIFAR-10 unlearned models (sample-wise unlearning) similar to Table 3.

| MIA Type | Rank 1 | | | Rank 2 | | |
|---|---|---|---|---|---|---|
| | **Full** | **Bottom 20%** | **Bottom 60%** | **Full** | **Bottom 20%** | **Bottom 60%** |
| Loss MIA | $0.504 \pm 0.011$ | $0.501 \pm 0.010$ | $0.502 \pm 0.007$ | $0.499 \pm 0.010$ | $0.509 \pm 0.009$ | $0.503 \pm 0.012$ |
| Conf MIA | $0.505 \pm 0.008$ | $0.496 \pm 0.005$ | $0.508 \pm 0.008$ | $0.499 \pm 0.008$ | $0.505 \pm 0.008$ | $0.499 \pm 0.008$ |
| Entropy MIA | $0.498 \pm 0.010$ | $0.490 \pm 0.009$ | $0.500 \pm 0.009$ | $0.496 \pm 0.007$ | $0.502 \pm 0.009$ | $0.494 \pm 0.010$ |
| Margin MIA | $0.502 \pm 0.009$ | $0.500 \pm 0.009$ | $0.514 \pm 0.007$ | $0.497 \pm 0.010$ | $0.501 \pm 0.011$ | $0.493 \pm 0.004$ |
| TopK(3) MIA | $0.505 \pm 0.007$ | $0.494 \pm 0.009$ | $0.499 \pm 0.010$ | $0.494 \pm 0.010$ | $0.502 \pm 0.009$ | $0.494 \pm 0.006$ |
| Combined MIA | $0.964 \pm 0.004$ | $0.959 \pm 0.003$ | $0.960 \pm 0.005$ | $0.966 \pm 0.003$ | $0.967 \pm 0.002$ | $0.966 \pm 0.003$ |

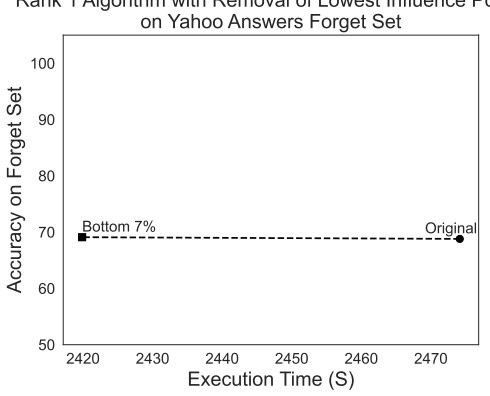

Figure I.6: Similarly to Figure 5.2 , we remove points from the forget set of Yahoo Answers and perform unlearning using the Rank 1 algorithm, and track changes in execution time (sample-wise unlearning). Execution time decreases with minimal changes to accuracy from the original unlearned model.

Table 5: **MIA tests for CIFAR-100 (logistic-regression):** Loss MIA, Conf MIA, Entropy MIA, Margin MIA, TopK (3) MIA, and Combined MIA values for the CIFAR-100 unlearned models (sample-wise unlearning) similar to Table 3.

| MIA Type | Rank 1 | | Rank 2 | |
|---|---|---|---|---|
| | **Full** | **Bottom 3%** | **Full** | **Bottom 3%** |
| Loss MIA | $0.589 \pm 0.009$ | $0.593 \pm 0.015$ | $0.537 \pm 0.007$ | $0.541 \pm 0.009$ |
| Conf MIA | $0.629 \pm 0.010$ | $0.646 \pm 0.015$ | $0.516 \pm 0.009$ | $0.512 \pm 0.013$ |
| Entropy MIA | $0.633 \pm 0.011$ | $0.645 \pm 0.015$ | $0.513 \pm 0.011$ | $0.515 \pm 0.013$ |
| Margin MIA | $0.625 \pm 0.011$ | $0.644 \pm 0.012$ | $0.512 \pm 0.008$ | $0.517 \pm 0.013$ |
| TopK(3) MIA | $0.642 \pm 0.007$ | $0.662 \pm 0.016$ | $0.512 \pm 0.009$ | $0.509 \pm 0.010$ |
| Combined MIA | $0.699 \pm 0.014$ | $0.698 \pm 0.015$ | $0.497 \pm 0.008$ | $0.500 \pm 0.011$ |

Table 6: **MIA tests for CIFAR-100 (Random Forest):** Loss MIA, Conf MIA, Entropy MIA, Margin MIA, TopK (3) MIA, and Combined MIA values for the CIFAR-100 unlearned models (sample-wise unlearning)

| MIA Type | Rank 1 | | Rank 2 | |
|---|---|---|---|---|
| | **Full** | **Bottom 3%** | **Full** | **Bottom 3%** |
| Loss MIA | $0.576 \pm 0.014$ | $0.571 \pm 0.010$ | $0.514 \pm 0.011$ | $0.503 \pm 0.007$ |
| Conf MIA | $0.533 \pm 0.010$ | $0.553 \pm 0.011$ | $0.518 \pm 0.008$ | $0.504 \pm 0.012$ |
| Entropy MIA | $0.541 \pm 0.014$ | $0.549 \pm 0.013$ | $0.494 \pm 0.014$ | $0.507 \pm 0.011$ |
| Margin MIA | $0.537 \pm 0.011$ | $0.539 \pm 0.013$ | $0.522 \pm 0.018$ | $0.501 \pm 0.018$ |
| TopK(3) MIA | $0.605 \pm 0.012$ | $0.611 \pm 0.011$ | $0.518 \pm 0.012$ | $0.498 \pm 0.011$ |
| Combined MIA | $0.948 \pm 0.003$ | $0.938 \pm 0.008$ | $0.948 \pm 0.004$ | $0.957 \pm 0.008$ |

Table 7: **MIA tests for Fashion-MNIST (Rank 1 Algorithm, logistic-regression):** Loss MIA, Conf MIA, Entropy MIA, Margin MIA, TopK (3) MIA, and Combined MIA values for the Fashion-MNIST unlearned models (sample-wise unlearning) similar to Table 3.

| MIA Type | Full | Bottom 17% |
|---|---|---|
| Loss MIA | $0.496 \pm 0.005$ | $0.497 \pm 0.007$ |
| Conf MIA | $0.495 \pm 0.007$ | $0.493 \pm 0.006$ |
| Entropy MIA | $0.493 \pm 0.007$ | $0.492 \pm 0.009$ |
| Margin MIA | $0.496 \pm 0.007$ | $0.491 \pm 0.005$ |
| TopK(3) MIA | $0.492 \pm 0.006$ | $0.489 \pm 0.006$ |
| Combined MIA | $0.493 \pm 0.004$ | $0.497 \pm 0.006$ |

Table 8: **MIA tests for Yahoo Answers (Rank 1 Algorithm, logistic-regression):** Loss MIA, Conf MIA, Entropy MIA, Margin MIA, TopK (3) MIA, and Combined MIA values for the Yahoo Answers unlearned models (sample-wise unlearning) similar to Table 3.

| MIA Type | Full | Forgetting 100000 |
|---|---|---|
| Loss MIA | $0.499 \pm 0.003$ | $0.497 \pm 0.002$ |
| Conf MIA | $0.498 \pm 0.002$ | $0.501 \pm 0.003$ |
| Entropy MIA | $0.498 \pm 0.002$ | $0.501 \pm 0.002$ |
| Margin MIA | $0.498 \pm 0.002$ | $0.501 \pm 0.002$ |
| TopK(3) MIA | $0.500 \pm 0.002$ | $0.501 \pm 0.002$ |
| Combined MIA | $0.501 \pm 0.003$ | $0.499 \pm 0.003$ |

Table 9: **Evaluation Metric using CIFAR-10 (Rank 1 Algorithm):** evaluation metric scores (forget and final score) of our unlearned models (using the Rank 1 algorithm, sample-wise unlearning, and N=20 model pairs). Standard errors are estimated via an unpaired bootstrap over model seeds (B = 200)

| Forgetting Type | Forget Score | Final Score (Utility-Adjusted) |
|---|---|---|
| Full Forgetting | $0.0191 \pm 0.0015$ | $0.0187 \pm 0.0015$ |
| Bottom 20% | $0.0197 \pm 0.0014$ | $0.0197 \pm 0.0014$ |
| Bottom 60% | $0.0213 \pm 0.0020$ | $0.0210 \pm 0.0017$ |

Table 10: **Evaluation Metric using CIFAR-100 (Rank 1 Algorithm):** evaluation metric scores (forget and final score) of our unlearned models (using the Rank 1 algorithm, sample-wise unlearning, and N=20 model pairs). similar to Table 9.)

| Forgetting Type | Forget Score | Final Score (Utility-Adjusted) |
|---|---|---|
| Full Forgetting | $0.0098 \pm 0.0023$ | $0.0107 \pm 0.0028$ |
| Bottom 3% | $0.0101 \pm 0.0024$ | $0.0111 \pm 0.0029$ |

Table 11: **Upper Bound:** Hessian (Test) influence calculation runtimes on full training datasets (forget + retain) using 1 H100 GPU.

| Dataset | Runtime (S) |
|---|---|
| CIFAR-10 | 25,537.09 |
| CIFAR-100 | 33,168.72 |
| Fashion-MNIST | 38,024.35 |
| Yahoo Answers Topic | 152,116.15 |

Table 12: **CIFAR-10 sample-wise unlearning:** Rank 1 algorithm with removal of lowest influence points (forget and retain) on CIFAR-10. Note: Bottom 60% on the forget set corresponds to Full retain (we remove points from both sets).

| Removal Target | Setting | Execution Time (S) | Accuracy (%) |
|---|---|---|---|
| Forget Set | Original | 60.66 | 92.97 |
| | Bottom 20% | 58.71 | 93.80 |
| | Bottom 40% | 55.31 | 93.57 |
| | Bottom 60% | 53.62 | 92.43 |
| Retain Set | Original | 60.66 | 97.93 |
| | Full retain | 53.62 | 99.37 |
| | Bottom 20% | 43.90 | 100.00 |
| | Bottom 40% | 34.36 | 99.90 |
| | Bottom 50% | 29.57 | 99.80 |

Table 13: **CIFAR-100 sample-wise unlearning**: Rank 1 and Rank 2 algorithms with removal of lowest influence points from both the forget and retain sets of CIFAR-100.

| Algorithm | Setting | Execution Time (S) | Retain Acc (%) | Forget Acc (%) |
|---|---|---|---|---|
| Rank 1 | Original | 61.51 | 100.00 | 81.37 |
| | Full retain and 7% forgetting | 60.53 | 99.97 | 81.03 |
| | Bottom 1% retain and 7% forgetting | 58.47 | 100.00 | 80.87 |
| Rank 2 | Original | 37.52 | 100.00 | 74.90 |
| | Full retain and 50% forgetting | 36.91 | 100.00 | 75.27 |
| | Bottom 4% retain and 50% forgetting | 35.42 | 99.80 | 74.60 |
| | Bottom 7% retain and 50% forgetting | 33.81 | 99.60 | 74.67 |

Table 14: Retrained accuracies on low influence points from the CIFAR-10 forget and retain sets when models are trained without access to them, compared to accuracies on randomly removed points.

| Removal Target | Proportion Removed | Low Influence Acc (%) | Random Acc (%) |
|---|---|---|---|
| Forget Set | Original (0.00) | 97.30 | – |
| | 0.10 | 99.20 | 91.70 |
| | 0.20 | 99.00 | 90.90 |
| | 0.40 | 98.60 | 92.30 |
| | 0.60 | 97.70 | 91.20 |
| Retain Set | Fully retrained (0.00) | 98.40 | – |
| | 0.10 | 99.00 | 90.60 |
| | 0.20 | 98.80 | 89.40 |
| | 0.40 | 95.50 | 89.00 |
| | 0.50 | 95.50 | 87.10 |
| | 0.60 | 90.90 | 86.10 |

Table 15: **CIFAR-10 sample-wise unlearning:** Rank 2 algorithm with removal of lowest influence points from CIFAR-10 forget and retain sets.

| Removal Target | Setting | Execution Time (S) | Accuracy (%) |
|---|---|---|---|
| Forget Set | Original | 24.04 | 92.77 |
| | Bottom 20% | 22.18 | 92.47 |
| | Bottom 40% | 22.05 | 92.20 |
| | Bottom 60% | 21.55 | 91.97 |
| Retain Set | Original | 24.04 | 100.00 |
| | Full retain | 21.55 | 100.00 |
| | Bottom 20% | 17.79 | 99.63 |
| | Bottom 40% | 13.43 | 98.40 |
| | Bottom 50% | 11.57 | 97.27 |

Table 16: **CIFAR-10 sample-wise unlearning:** Rank 3 algorithm with removal of lowest influence points on the CIFAR-10 retain set.

| Setting | Execution time (S) | Accuracy on original retain set (%) |
|---|---|---|
| Original | 212.32 | 100.00 |
| Bottom 20% | 172.03 | 99.70 |
| Bottom 40% | 131.88 | 98.93 |
| Bottom 50% | 110.44 | 98.17 |

Table 17: **Fashion-MNIST sample-wise unlearning:** Rank 3 and Rank 1 algorithms with removal of lowest influence points from the Fashion-MNIST forget set.

| Rank | Setting | Execution Time (S) | Accuracy on Forget Set (%) |
|---|---|---|---|
| Rank 3 | Original | 868.86 | 90.00 |
| | Bottom 8% | 868.26 | 90.00 |
| | Bottom 17% | 865.50 | 89.20 |
| Rank 1 | Original | 1085.80 | 90.50 |
| | Bottom 17% | 1052.88 | 91.20 |

Table 18: **Yahoo Answers sample-wise unlearning:** Rank 1 algorithm with removal of lowest influence points from the Yahoo Answers forget set.

| Setting | Execution time (S) | Accuracy on forget set (%) |
|---|---|---|
| Original | 2474.15 | 68.80 |
| Bottom 7% | 2419.87 | 69.10 |

Table 19: **CIFAR-10 class-wise unlearning:** Accuracy changes when classes 0, 2, and 5 are removed from CIFAR-10 using the Rank 1 algorithm.

| Class | Forgetting Type | Test Acc (%) | Forget Acc (%) | Retain Acc (%) | Execution Time (S) |
|---|---|---|---|---|---|
| 0 | Full Forgetting | 75.1 | 0.0 | 92.2 | 141.26 |
|  | Bottom 20% | 73.3 | 0.0 | 88.4 | 132.98 |
|  | Bottom 40% | 74.4 | 0.0 | 89.4 | 127.40 |
|  | Bottom 60% | 74.9 | 0.0 | 91.9 | 119.97 |
| 2 | Full Forgetting | 76.1 | 0.0 | 91.3 | 146.47 |
|  | Bottom 20% | 76.7 | 0.0 | 91.6 | 136.31 |
|  | Bottom 40% | 74.5 | 0.0 | 87.3 | 130.47 |
|  | Bottom 60% | 75.5 | 0.0 | 88.5 | 125.22 |
| 5 | Full Forgetting | 75.1 | 0.0 | 89.2 | 145.84 |
|  | Bottom 20% | 74.4 | 0.0 | 89.5 | 137.99 |
|  | Bottom 40% | 78.3 | 0.0 | 94.9 | 132.26 |
|  | Bottom 60% | 77.5 | 0.0 | 94.4 | 127.58 |
| **Average (Full Forgetting)** | | 75.4 | 0.0 | 90.9 | 144.52 |
| **Average (Bottom 20%)** | | 74.8 | 0.0 | 89.8 | 135.76 |
| **Average (Bottom 40%)** | | 75.7 | 0.0 | 90.5 | 130.04 |
| **Average (Bottom 60%)** | | 76.0 | 0.0 | 91.6 | 124.26 |

Table 20: MIA values (logistic-regression) for CIFAR-10 class-wise removal on classes 0, 2, and 5 using the Rank 1 algorithm.

| Class | Forgetting Type | Loss MIA | Conf MIA | Entropy MIA | Margin MIA |
|---|---|---|---|---|---|
| 0 | Full Forgetting | $0.942 \pm 0.004$ | $0.682 \pm 0.013$ | $0.714 \pm 0.012$ | $0.684 \pm 0.008$ |
| | Bottom 20% | $0.947 \pm 0.004$ | $0.653 \pm 0.008$ | $0.666 \pm 0.010$ | $0.640 \pm 0.009$ |
| | Bottom 40% | $0.943 \pm 0.003$ | $0.679 \pm 0.007$ | $0.698 \pm 0.005$ | $0.670 \pm 0.011$ |
| | Bottom 60% | $0.943 \pm 0.004$ | $0.645 \pm 0.012$ | $0.668 \pm 0.009$ | $0.643 \pm 0.008$ |
| 2 | Full Forgetting | $0.946 \pm 0.003$ | $0.696 \pm 0.007$ | $0.723 \pm 0.007$ | $0.684 \pm 0.008$ |
| | Bottom 20% | $0.947 \pm 0.003$ | $0.669 \pm 0.010$ | $0.685 \pm 0.007$ | $0.665 \pm 0.008$ |
| | Bottom 40% | $0.944 \pm 0.003$ | $0.643 \pm 0.009$ | $0.665 \pm 0.008$ | $0.644 \pm 0.008$ |
| | Bottom 60% | $0.948 \pm 0.003$ | $0.680 \pm 0.010$ | $0.694 \pm 0.009$ | $0.670 \pm 0.010$ |
| 5 | Full Forgetting | $0.951 \pm 0.004$ | $0.642 \pm 0.008$ | $0.647 \pm 0.008$ | $0.630 \pm 0.007$ |
| | Bottom 20% | $0.944 \pm 0.004$ | $0.546 \pm 0.011$ | $0.550 \pm 0.012$ | $0.535 \pm 0.012$ |
| | Bottom 40% | $0.948 \pm 0.004$ | $0.590 \pm 0.007$ | $0.594 \pm 0.010$ | $0.584 \pm 0.010$ |
| | Bottom 60% | $0.948 \pm 0.003$ | $0.556 \pm 0.007$ | $0.552 \pm 0.007$ | $0.555 \pm 0.005$ |
| **Average (Full Forgetting)** | | $0.946 \pm 0.002$ | $0.673 \pm 0.006$ | $0.695 \pm 0.005$ | $0.666 \pm 0.004$ |
| **Average (Bottom 20%** | | $0.946 \pm 0.002$ | $0.623 \pm 0.006$ | $0.634 \pm 0.006$ | $0.613 \pm 0.006$ |
| **Average (Bottom 40%** | | $0.945 \pm 0.002$ | $0.637 \pm 0.004$ | $0.652 \pm 0.005$ | $0.633 \pm 0.006$ |
| **Average (Bottom 60%** | | $0.946 \pm 0.002$ | $0.627 \pm 0.006$ | $0.638 \pm 0.005$ | $0.623 \pm 0.005$ |

| Class | Forgetting Type | TopK(3) MIA | Combined MIA |
|---|---|---|---|
| 0 | Full Forgetting | $0.700 \pm 0.012$ | $0.941 \pm 0.004$ |
| | Bottom 20% | $0.669 \pm 0.008$ | $0.946 \pm 0.004$ |
| | Bottom 40% | $0.685 \pm 0.007$ | $0.942 \pm 0.003$ |
| | Bottom 60% | $0.660 \pm 0.011$ | $0.940 \pm 0.004$ |
| 2 | Full Forgetting | $0.706 \pm 0.005$ | $0.945 \pm 0.003$ |
| | Bottom 20% | $0.685 \pm 0.009$ | $0.945 \pm 0.004$ |
| | Bottom 40% | $0.662 \pm 0.009$ | $0.942 \pm 0.003$ |
| | Bottom 60% | $0.689 \pm 0.008$ | $0.947 \pm 0.003$ |
| 5 | Full Forgetting | $0.648 \pm 0.007$ | $0.949 \pm 0.004$ |
| | Bottom 20% | $0.537 \pm 0.007$ | $0.943 \pm 0.004$ |
| | Bottom 40% | $0.592 \pm 0.009$ | $0.945 \pm 0.005$ |
| | Bottom 60% | $0.554 \pm 0.008$ | $0.945 \pm 0.004$ |
| **Average (Full Forgetting)** | | $0.685 \pm 0.005$ | $0.945 \pm 0.002$ |
| **Average (Bottom 20%** | | $0.630 \pm 0.005$ | $0.945 \pm 0.002$ |
| **Average (Bottom 40%** | | $0.646 \pm 0.005$ | $0.943 \pm 0.002$ |
| **Average (Bottom 60%** | | $0.634 \pm 0.005$ | $0.944 \pm 0.002$ |

Table 21: **CIFAR-100 subclass-wise unlearning:** Accuracy changes when subclasses 1, 10, and 16 from CIFAR-100 are removed using the Rank 1 algorithm.

| Class | Forgetting Type | Test Acc (%) | Forget Acc (%) | Retain Acc (%) | Execution Time (S) |
|---|---|---|---|---|---|
| 1 | Full Forgetting | 77.6 | 0.0 | 100.0 | 81.93 |
| | Bottom 10% | 78.5 | 0.0 | 100.0 | 81.16 |
| | Bottom 20% | 77.3 | 0.0 | 100.0 | 79.53 |
| | Bottom 40% | 78.2 | 0.0 | 100.0 | 78.77 |
| 10 | Full Forgetting | 79.3 | 0.2 | 100.0 | 54.93 |
| | Bottom 10% | 79.6 | 0.0 | 100.0 | 54.79 |
| | Bottom 20% | 79.3 | 0.0 | 100.0 | 53.63 |
| | Bottom 40% | 79.7 | 0.0 | 100.0 | 51.71 |
| 16 | Full Forgetting | 78.1 | 0.0 | 100.0 | 80.92 |
| | Bottom 10% | 77.4 | 0.0 | 100.0 | 82.09 |
| | Bottom 20% | 77.8 | 0.0 | 100.0 | 81.70 |
| | Bottom 40% | 78.3 | 0.0 | 100.0 | 79.65 |
| **Average (Full Forgetting)** | | 78.3 | 0.1 | 100.0 | 72.59 |
| **Average (Bottom 10%)** | | 78.5 | 0.0 | 100.0 | 72.68 |
| **Average (Bottom 20%)** | | 78.1 | 0.0 | 100.0 | 71.62 |
| **Average (Bottom 40%)** | | 78.7 | 0.0 | 100.0 | 70.04 |

## J   Lowest gradients analysis

As we mention, training order can affect point influence scores, because model learning is more likely to be affected by initial exposures to a certain class or output (Appendix E). As training continues, points similar to earlier ones typically impact learning less. To best understand when we should calculate our Lowest Gradients method to find points that are initially minimally impactful, we also explore the distribution of low gradients during checkpoints of training. In Figure J.1, we measure the quantity of training points who exhibit a minimal L2 gradient, and thus model impact, over the course of training. As the model fits to the training points, single point contributions diminish and the total number of low gradient points increases. This fitting is likely occurring at approximately Checkpoint 5, as the number of training points with a low gradient rapidly rises. However, as can be observed, before Checkpoint 5, there is a steady number of low gradient points that slowly increases. These points make a minimal impact to model learning from the beginning of training, and as such draw a distinction between points that could be impactful dependent on factors like training regime order, and points that are minimally impactful from the beginning.

## K   Hyperparameters

### K.1   Influence calculations

When training ResNet-101 models, we use a common learning rate parameter of "lr"=0.003, a momentum parameter of "momentum"=0.9, and a batch size of 128. For the purposes of the Lowest Gradients influence estimation, as described in Section 3, we identify datapoints that achieve a low $L2$ norm after checkpoint 5 (out of 12), and remain low for the duration of training. In this instance, we treat all datapoints in the lowest 5% of the training set as low influence. For all datasets, we use the standardly prescribed training, testing, and validation dataset splits.

### K.2   Unlearning competition algorithms

Across all our experiments, we randomly select the seed 42, and use it consistently to ensure reproducibility.

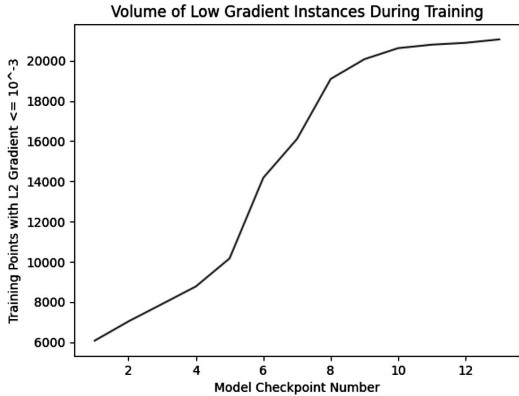

Figure J.1: Throughout training on CIFAR-10 (x-axis), we track the number of training points that exhibit a minimal L2 gradient (y-axis). As can be seen, around Checkpoint 5, the number of low L2 gradient points increases significantly, implying that model fitting is occurring more rapidly. Training points that consistently have a low L2 gradient across checkpoints may be correlated with low influence as well.

**CIFAR-10**: We use the top 3 ranked algorithms from the NeurIPS'23 competition on unlearning to evaluate our unlearning regime. In order to do this, we first run hyperparameter sweeps on both algorithms given our recreated image classification model. By running these sweeps, we aim to optimize the values of the unlearned model such that the accuracy of the retain set matches the original training (indicating not forgetting) and the forget set matches the test set (indicating forgetting). Specifically, for the 1st ranked algorithm, we run a standard sweep on "epochs" and the learning rate of "optimizer_forget" and find the optimal values to be 10 and 9e-3 respectively. For the 2nd ranked algorithm, we find the optimal values for learning rate "lr"=0.003 and "epoch"=6. For the 3rd ranked algorithm, we run a standard sweep on "epochs" and the learning rate of "optimizer" and find that the optimal values are 60 and 0.1. To compute accuracies, we run each algorithm n=3 and average results.

**CIFAR-100**: Following from above, we use the same top 3 ranked algorithms and run hyperparameter sweeps given our fine-tuned model on CIFAR-100. We find the following hyperparameters to be optimal: "epochs"=20, the "lr" of "optimizer" to be 1e-5, the "lr" of "optimizer_forget" to be 9e-5 for the 1st ranked algorithm, "lr"=0.05 and "epoch"=20 for the 2nd ranked, and "epochs"=30 with "lr" of "optimizer" to be 0.015 for the 3rd ranked. The 3rd ranked algorithm also hard-codes for 10 classes. To accommodate our 50-class subset of CIFAR-100, we use a vanilla cross-entropy without reweighting Triantafillou et al. (2024). To compute accuracies, we run each algorithm n=3 and average results.

**Fashion-MNIST**: Following from above, we use the Rank 1 and Rank 3 algorithms and run hyperparameter sweeps given our fine-tuned model on Fashion-MNIST. We find the following hyperparameters to be optimal: "epochs"=1, the "lr" of "optimizer" to be 5e-3, the "lr" of "optimizer_forget" to be 9e-3, "retain_bs" to be 64, for the 1st ranked algorithm, and "epochs"=1 with "lr" of "optimizer" to be 0.1 for the 3rd ranked.

**Yahoo Answers**: Following from above, we use the Rank 1 algorithm, as it is the only effective algorithm for this language classification task. We run hyperparameter sweeps given our fine-tuned model on Yahoo Answers. We find the following hyperparameters to be optimal: "epochs"= 1, the "lr" of "optimizer" to be 5e-3, the "lr" of "optimizer_forget" to be 9e-3, "retain_bs" to be 64.

### K.3 Ablation study

We perform an ablation study on the hyperparameters we tune in our three unlearning algorithms (Tables 22 to 24). We set up a class-wise unlearning scenario using Class 1 of CIFAR-10, before running each unlearning algorithm on the full forget set. For the Rank 1 algorithm, we vary the number of epochs the algorithm trains

on the retain set, as well as, the retain loader batch size. For Rank 2 and Rank 3, we vary the learning rate used during training on the retain set, as well as, number of epochs over the retain loader. Across examples, we find that increasing number of epochs $e$ generally improves performance across all algorithms until convergence. Furthermore, increasing learning rate $lr$ improves performance up to a point. Finally, moderately sized batches yield the best performance-runtime tradeoff.

Table 22: Rank 1 algorithm ablation study on CIFAR-10 (Class 0, Full Forgetting): effect of epochs ($e$) and retain batch size ($r$) on performance and execution time.

| $e$ | $r$ | Execution Time (S) | Test Acc (%) | Forget Acc (%) | Retain Acc (%) |
|---|---|---|---|---|---|
| 10 | 256 | 70.39 | 67.5 | 0.0 | 77.1 |
| 20 | 64 | 171.29 | 74.2 | 0.0 | 89.9 |
| 20 | 128 | 141.26 | 75.1 | 0.0 | 92.2 |
| 20 | 256 | 123.98 | 75.0 | 0.0 | 90.5 |
| 20 | 500 | 118.48 | 70.5 | 0.0 | 82.2 |
| 30 | 256 | 123.98 | 75.0 | 0.0 | 90.5 |

Table 23: Rank 2 algorithm ablation study on CIFAR-10 (Class 0, Full Forgetting): effect of learning rate ($lr$) and epochs ($e$) on performance and execution time.

| $e$ | $lr$ | Execution Time (S) | Test Acc (%) | Forget Acc (%) | Retain Acc (%) |
|---|---|---|---|---|---|
| 1 | 0.01 | 13.87 | 76.8 | 0.1 | 92.2 |
| 6 | 0.001 | 43.87 | 78.5 | 3.8 | 98.8 |
| 6 | 0.008 | 70.51 | 81.6 | 0.0 | 100.0 |
| 6 | 0.01 | 69.00 | 82.7 | 0.0 | 100.0 |
| 6 | 0.1 | 68.14 | 82.3 | 0.0 | 99.9 |
| 10 | 0.01 | 72.34 | 82.8 | 0.0 | 100.0 |

Table 24: Rank 3 algorithm ablation study on CIFAR-10 (Class 0, Full Forgetting): effect of epochs ($e$) and learning rate ($lr$) on performance and execution time.

| $e$ | $lr$ | Execution Time (S) | Test Acc (%) | Forget Acc (%) | Retain Acc (%) |
|---|---|---|---|---|---|
| 2 | 0.1 | 15.24 | 90.6 | 66.2 | 100.0 |
| 6 | 0.01 | 44.90 | 91.7 | 91.0 | 100.0 |
| 6 | 0.1 | 44.94 | 85.1 | 4.2 | 100.0 |
| 6 | 0.5 | 45.05 | 82.7 | 0.0 | 99.7 |
| 10 | 0.1 | 110.79 | 84.9 | 0.0 | 100.0 |
| 30 | 0.1 | 327.98 | 84.3 | 0.0 | 100.0 |

## L  Computing environment

We train our models and run the unlearning algorithms using 1 NVIDIA H100 or A100 GPU and request a maximum of 700G for memory on Linux systems. We train our models and load our datasets using PyTorch (Paszke et al., 2019). To compute influence scores using Hessian approximation, we use the implementation from nimarb/pytorch_influence_functions[12] substituted with our datasets and models. For LESS, we use the implementation from princeton-nlp/LESS[13]. Finally, for calculating Lowest Gradients, we implement our own solution to identify points in the training set with low gradient norms early on in training as described. To run the unlearning algorithms, we base our setup on the starting kit from the Unlearning

---

[12]https://github.com/nimarb/pytorch_influence_functions
[13]https://github.com/princeton-nlp/LESS

Competition[14]. We modify the initial model used, as well as the data fed to each algorithm accounting for removed points.

