# OpenReview forum: "When unlearning is free: leveraging low influence points to reduce computational costs"
_TMLR — Rejected by TMLR_

### Review · Reviewer_Ec8b · 2025-12-30

**Summary Of Contributions:**

This paper argues that not all points in the forget set matter equally for unlearning, and that a sizable fraction of requested deletions may be “low-impact” in the sense that removing them would barely change model behavior. The authors study influence-function–style approximations (Hessian-based influence, LESS, and a “Lowest Gradients” heuristic) to identify a subset of low-influence points and show empirically that models retrained without these points still generalize well to them (i.e., accuracy on the removed points stays high up to a threshold).  Building on this, they propose an framework: compute influence scores once, select the bottom lowest-influence points, and then remove their intersection from the forget set and (optionally) the retain set before running an existing unlearning method. In a recreated NeurIPS’23 unlearning-competition setup (CIFAR-10 and a CIFAR-100 subset), combining this filtering with top competition methods yields up to ~50% execution-time reduction while maintaining similar retain/forget accuracy and similar [the simple version] MIA accuracy.

**Audience:**

No

**Audience Explanation:**

Without rigorous evaluations (more models/datasets, stronger metrics, and further baselines), I don't think the results would be interesting and reliable to report.

**Claims And Evidence:**

No

**Claims Explanation:**

Strength:

S1: The core premise that not all data points require equal computational effort to unlearn is intuitive and practically valuable.

S2: The unlearning integration is method-agnostic and easily usable with any method.

S3: The experiments span two data modalities and tasks, i.e., vision and language.



Weaknesses:

W1. This work's contribution is mostly empirical, but the empirical evaluation is limited. The study uses essentially one language dataset/model (SQuAD + BERT embeddings) and the end-to-end unlearning evaluation is only on CIFAR-style vision (CIFAR-10 + a CIFAR-100 subset) with a specific ResNet-18 competition setup. This limits conclusions about generality (other datasets, other architectures, other unlearning pipelines).


W2. the MIA used is a logistic regression attack from the competition starter kit, which is not a strong adversary by modern standards, and it is the only evaluation for the argument that privacy guarantees carry over. he NeurIPS’23 competition emphasized a more rigorous evaluation score, but the paper does not use that and instead adopts the starter-kit metrics “for consistency,” which is not fully satisfying.


W3. the studied baselines are limited to the top-3 competition methods, which are a very specific family of heuristic unlearning approaches.


W4. the paper does not give concrete runtime or memory numbers for influence computation itself, making it hard to assess the true end-to-end savings, especially when influence must be computed post hoc.


W5. in Figure 5.1, why is the accuracy on the forget-set so high? Does it mean the models are not forgetting at all?


W6. There is no real scalability study of the framework (e.g., larger models, larger datasets, more clients/requests, repeated deletions).

W7. The presentation of the results can be improved. for instance, Table 1 and Table 2 miss borders and structure, making them difficult to read. Also, the overlapping or statistically similar results should be formatted (e.g., bolded) to help readability and interpretation.

**Requested Changes:**

Please address the weaknesses raised above.

---

> ### Author Response · Authors · 2026-02-24
>
> We thank the reviewer for their thoughtful and detailed feedback. In response, we have substantially expanded our empirical evaluation (changes are highlighted in the manuscript). To summarize, we have introduced experiments on two new unlearning pipelines (class-wise and subclass-wise, in addition to our original sample-wise setting), as well as, have added an additional language and additional vision model to our evaluations. Our evaluations themselves now feature a suite of six MIAs using both linear and nonlinear models, as well as, the evaluation of our models on the competition’s official metric (https://unlearning-challenge.github.io/assets/data/Machine_Unlearning_Metric.pdf), alongside our original accuracy on the retain, forget and test sets. We also now include a comprehensive cost analysis that accounts for influence computation. Finally, we have also added an ablation study on the hyperparameters of our unlearning algorithms. We address the requested changes in order below:
>
>
> 1) We have both expanded our pipeline assessment to include class-wise and subclass-wise forget sets (in addition to our original sample-wise), as well as, have added 2 additional data/model settings: DistilBERT trained on Yahoo Answers, and ViT trained on Fashion-MNIST (in addition to our existing results on CIFAR-10, CIFAR-100 and SQuAD).
> We believe these enhancements more closely align with other unlearning papers and allow for better conclusions about generality.
>
> 2) We have expanded our metrics significantly: we now test on a suite of six MIAs that use different features, as well as, both a linear and non-linear attack model. We have also implemented two versions of the official competition evaluation on different datasets. Finally, we also continue to report accuracy on the forget, retain, and test sets following unlearning, and unlearning execution times. We believe this larger group of metrics allows for more comprehensive and reliable evaluation.
>
> 3) We selected the top-3 competition methods because they were the best performing, regardless of their specific family. Furthermore, their inherent treatments of the forget and retain sets, as well as, general approaches differ substantially, therefore we believe it is useful to compare performance differences among them. However, we would be happy to include additional baselines if the reviewer feels they would strengthen the evaluation..
>
> 4) We have added a more thorough cost analysis which involves explicit runtimes to calculate influence by dataset. Specifically, we offer an upper and lower bound of influence calculations given our influence approximation methods, to allow users to better select between methods.
>
> 5) Regarding Fig. 5.1: the sample-wise forget set spans multiple classes and remains interspersed with highly similar retained examples, making complete forgetting inherently more difficult. This leads to the observed ~8% drop in forget-set accuracy (compared to the model before unlearning) before test-set degradation occurs. In contrast, in class-wise and subclass-wise settings, forget accuracy reaches 0% (Table 18), reflecting the differences between these unlearning settings. We have clarified this distinction in the figure caption.
>
> 6) We have greatly expanded our evaluation to include larger and smaller datasets, both vision and language models, different types of unlearning pipelines (sample vs class vs subclass) with larger/smaller requests as mentioned above.
>
> 7) We have made the requested format changes, as well as, generally polished the manuscript and made other revisions to enhance readability.
>
> We once again thank the reviewer for their constructive feedback. We are happy to make any additional changes that would further strengthen the paper.

---

### Review · Reviewer_e4gD · 2026-02-02

**Summary Of Contributions:**

This paper argues that a lot of “unlearning work” is spent on training points that barely matter. The authors propose a simple wrapper: estimate influence, throw away the bottom x% of low-impact points from the forget/retain sets, and then run any standard unlearning method on the remaining high-impact subset. In experiments, this trims unlearning runtime substantially (they report up to ~50% reduction in some settings) while keeping utility and their membership-inference proxy roughly stable.

**Audience:**

Yes

**Audience Explanation:**

Yes , I think at least some people in TMLR’s audience would care about this paper’s findings.

A lot of TMLR readers work on practical learning systems (optimization, robustness, privacy, data quality, efficiency). This paper sits right in that space: it says you can often cut unlearning compute by filtering out low-impact points using influence-style scores, without materially changing utility or their membership-inference proxy in the tested setups. That’s a pretty actionable message for anyone building or evaluating unlearning pipelines, especially because it’s presented as a wrapper you can attach to existing methods rather than a totally new algorithm.

**Broader Impact Concerns:**

- Privacy evaluation feels a bit narrow. The membership inference check is helpful as a sanity test, but it’s not the strongest adversary, and it’s not a formal unlearning guarantee. I’d be cautious about overstating privacy conclusions based mainly on this metric.


- Theory is more motivational than predictive. The theoretical framing relies on assumptions (e.g., convexity/unique minimizers) that don’t match modern deep nets, so it reads more like intuition-building than a real explanation of when it will or won’t work.


- End-to-end cost accounting could be clearer. Influence estimation itself can be non-trivial overhead (they discuss post-training cost), so I’d like to see more emphasis on net savings, especially when influence can’t be amortized across many unlearning requests.


- Risky corner cases are exactly the ones I worry about for unlearning. The duplicates/group-influence issue isn’t hypothetical; real-world data has templates, repeated patterns, and near-duplicates. This is the kind of case where “ignore low-influence points” could backfire in a way that’s hard to detect.


- Scale gap to the headline motivation. CIFAR and the language experiment are reasonable research benchmarks, but the motivating pressure for unlearning is increasingly at much larger scale. I’m not fully convinced the same “bottom x% is safe to ignore” behavior will transfer without surprises.

**Claims And Evidence:**

Yes

**Claims Explanation:**

- Practical, easy-to-understand idea. The framework is simple to implement and doesn’t require inventing a whole new unlearning algorithm—more like a speed/efficiency “front-end” you can attach to existing methods.


- Method-agnostic and tested with strong baselines. They plug the approach into competitive unlearning challenge methods, which makes the contribution feel more grounded than a single bespoke pipeline.


- Comparative look at influence estimators. I appreciated that they didn’t treat “influence” as one monolithic thing—there’s a real comparison between Hessian-based approximations, LESS, and a cheap gradient heuristic, and they report that self-influence variants tend to be strongest.


- Cross-domain evidence. It’s not just one dataset: they show vision (CIFAR-10/100) and a language setting (SQuAD/BERT-style setup), which helps with confidence that the effect isn’t totally brittle.

**Requested Changes:**

- Add at least one stronger privacy stress test (multiple MIAs, stronger features/attack models, maybe a white-box variant where applicable).

- Report net runtime explicitly (influence computation + unlearning), and discuss when influence can be reused across many deletion requests

- Provide a simple practical guideline for choosing x (even a heuristic based on the influence curve’s “knee” would help).

---

> ### Author Response · Authors · 2026-02-24
>
> We sincerely thank the reviewer for their thoughtful and constructive comments, as well as for raising broader concerns that we will continue to reflect on as we develop this work. Below, we address the requested changes (all updates are highlighted in the manuscript):
>
> 1) We now include a suite of six membership inference attacks (MIAs) based on different feature sets, evaluated using both linear and nonlinear attack models. In addition, we implemented two variations of the official competition evaluation metric (https://unlearning-challenge.github.io/assets/data/Machine_Unlearning_Metric.pdf). We continue to report accuracy on the retain, forget, and test sets. We believe these expanded evaluations provide a more comprehensive assessment of privacy and utility.
>
> 2) We have added a more detailed cost analysis that includes explicit runtimes for influence computation. Specifically, we provide empirical upper and lower bounds based on our least and most computationally intensive influence approximation methods, allowing users to better understand the range of preprocessing costs. We present this alongside a broader discussion of trade-offs to help clarify potential advantages and limitations under different practical settings.
>
> 3) We have also added a practical guideline and recommendation for selecting x based on our empirical findings.
>
> Beyond these revisions, we substantially expanded our empirical evaluation by adding experiments on two additional model-dataset pairs and incorporating two new unlearning pipelines (class-wise and subclass-wise, in addition to sample-wise). Finally, we included an ablation study on the hyperparameters of our unlearning algorithms. We believe these additions strengthen the generalizability of our results and broaden the applicability of our framework.
>
> We would be happy to implement any further changes the reviewer believes would enhance the paper.

---

> > ### Comment · Reviewer_e4gD · 2026-04-05
> >
> > Thank you for addressing these points. I have carefully reviewed the revisions, and the changes are clear and well presented. The responses adequately resolve the concerns I raised. From my perspective, everything looks good, and I have no further points to raise.

---

### Review · Reviewer_4f5t · 2026-02-20

**Summary Of Contributions:**

This paper proposed a method to filter out low-influence datapoints from the unlearn set before conducting unlearning. The paper experimented with three top-ranked unlearning algorithms from the 2023 NeurIPS Unlearning challenge.

**Audience:**

Yes

**Audience Explanation:**

Unlearning is a very interesting topic and would definitely gather the community of TMLR.

I, for one, would love to see high-quality works in this area!

**Claims And Evidence:**

No

**Claims Explanation:**

The experiment setup and and benchmarks used in this work is rather unconventional in the line of unlearning works. Typically, you would have to include sample-wise and class-wise removal. The ablation study with different hyperparameters is also missing.

There is also insufficient evidence to answer the main research question presented in this paper. More experiments are needed to convince me that removing a portion of the unlearning set will not significantly decrease the unlearning quality. Explained more in the section below.

**Requested Changes:**

First and foremost, I encourage the authors to thoroughly revise and polish the manuscript. There are multiple instances of vague, imprecise, or informal phrasing (i.e., “fully trained CIFAR-10 model”) that weaken the technical presentation.

The authors are recommended to review recent works in the line of machine unlearning to understand the general experiment setup and benchmark methods, such as [1]. The current paper appears to evaluate essentially one setup derived from the 2023 Unlearning Competition. This is insufficient to answer the core question posed in the paper. The ablation study with different hyperparameters is also missing: it was mentioned in the paper they did experiment with different set of parameters, it would be interesting to see those results.

More experimental data is needed to answer the main question presented in the paper. For one, I would want to see the unlearning efficacy with / without the identified low-influence points. The current paper appears to evaluate essentially one setup derived from the 2023 NeurIPS Unlearning Competition. This is insufficient to answer the core question posed in the paper..

The paper could also benefit from the full set of experimental parameters. Providing full experimental details would improve reproducibility.

Some figures presented in the paper are difficult to interpret. For example, Figures 5.1 and 5.2 appear nearly horizontal across varying execution times. It seems like the forget set accuracy remained high, but in most recent works the unlearning accuracy approached 0. What caused the difference? Presenting results in tabular form and improving figure captions would greatly improve the clarity.

The unlearning framework (presented in step 1 - 5) would benefit from being presented as formal pseudocode. This would improve clarity and help readers understand implementation details.

[1] Xindi Fan, Jing Wu, Mingyi Zhou, Pengwei Liang, & Dinh Phung. (2025). IMU: Influence-guided Machine Unlearning.

---

> ### Author Response · Authors · 2026-02-24
>
> We thank the reviewer for their thoughtful feedback. In response, we have substantially expanded our empirical evaluation (all changes are highlighted in the manuscript). In summary, we introduced experiments on two additional unlearning pipelines (class-wise and subclass-wise, in addition to sample-wise) and added new language and vision model-dataset pairs. Our evaluation now includes a suite of six MIAs using both linear and nonlinear attack models, two variations of the official competition evaluation metric (https://unlearning-challenge.github.io/assets/data/Machine_Unlearning_Metric.pdf), and accuracy on the retain, forget, and test sets, along with execution time. We also added a comprehensive cost analysis that accounts for influence computation prior to unlearning, and an ablation study on the hyperparameters of our unlearning algorithms. We address the reviewer’s specific points below:
>
> 1) We carefully revised the manuscript to eliminate imprecise phrasing and improve clarity. Please let us know if any issues remain.
>
> 2) We reviewed recent work on machine unlearning, including [1], and expanded our empirical evaluation accordingly. We now consider three unlearning setups: sample-wise, class-wise, and subclass-wise (inspired by [1]), to better align with standard evaluation practices. We also incorporated two additional dataset-model pairs to strengthen generalization. Our metrics were significantly broadened: we now report six MIAs (with linear and nonlinear attacks), two variations of the official Unlearning Evaluation Metric (https://unlearning-challenge.github.io/assets/data/Machine_Unlearning_Metric.pdf), and accuracy on the retain, forget, and test sets. We believe these additions provide a more rigorous and literature-aligned evaluation.
>
> 3) We expanded our empirical comparisons and evaluations on unlearning performance with and without low-influence point removal (see Figs. 5.1, 5.2, H.2, H.3, H.5, H.6).
>
> 4) We provide full experimental details in Appendix J to support reproducibility and include supplementary code demonstrating our framework on an example.
>
> 5) We added tabular results corresponding to all figures and expanded figure captions.
>
> 6) The high forget-set accuracy in Fig. 5.1 arises from the structure of the forget set. In the class-wise and subclass-wise settings, forget accuracy drops to 0 (Tables 19 and 21). However, in Fig. 5.1 the forget set spans multiple classes and remains interspersed with similar retained examples, making complete forgetting inherently more difficult.
>
> 7) We have rewritten our framework using pseudocode.
>
> We again thank the reviewer for their thoughtful feedback and would be happy to incorporate any further suggestions that would strengthen the manuscript or improve its alignment with best practices in machine unlearning.

---

### Author Response · Authors · 2026-03-07

We sincerely thank the reviewers for the detailed and constructive feedback. In our revision, we have made substantial additions to the experimental scope, metrics, and presentation (all highlighted in the manuscript). In particular, we made the following changes to address each specified weakness:

-Weakness: The evaluation is too narrow, limiting generality to other unlearning settings and models/datasets (Reviewers 4f5t, Ec8b): We expanded our framework to now evaluate on three unlearning settings (class-wise, subclass-wise, in addition to our existing sample-wise) and added another vision and another language model/dataset pair to test robustness beyond the original CIFAR/SQuAD models.


-Weakness: Privacy testing uses only the single starter-kit MIA (Reviewers e4gD, Ec8b): We now report results on a suite of six MIAs using both linear and non-linear attack models, as well as, two variants of the official competition unlearning evaluation metric, alongside existing retain/forget/test set accuracy and execution time.


-Weakness: The cost does not take into account influence calculation overhead (Reviewers e4gD, Ec8b): We added a comprehensive cost analysis reporting influence computation times (upper/lower bounds across methods), in addition to our existing unlearning execution times.


-Weakness: Some figures are hard to interpret (Reviewers 4f5t, Ec8b): We added tabular counterparts to our main figures, improved formatting/captions, and clarified why forget accuracy can remain high in the sample-wise case while reaching ~0 in class/subclass settings.


-Weakness: The hyperparameter ablation study is missing (Reviewer 4f5t): We added hyperparameter sweeps across our unlearning methods, more experimental setting details, and a clearer, more reproducible description of the overall framework.


-Weakness: There is no practical guidance for choosing the filtering threshold (x) (Reviewer e4gD): We added an empirical guideline for selecting (x) (with a recommended starting point and how to adjust).

We hope these revisions directly address the main weaknesses raised across reviews and strengthen the paper. If any concerns remain, we would be happy to incorporate further changes.

---

### Decision · Action_Editor_W6iM · 2026-07-01

**Recommendation:** Reject

**Additional Comments:**

Based on the reviews, it is clear that the paper requires a major revision to make sure it meets the requirement that all claims made in the submission supported by accurate, convincing and clear evidence.

Following TMLR policy, this can in most cases be achieved both by providing better evidence and by adjusting the claims to match the existing evidence.

While preparing the revision, I remind the authors that TMLR does not have a strict 12 page limit, so you are allowed to add additional material to the main text, as long as its inclusion and the length of the paper are justifiable.

**Audience:**

Yes

**Audience Explanation:**

All reviewers agree that the findings would be interesting to at least some individuals in TMLR's audience.

**Claims And Evidence:**

No

**Claims Explanation:**

I concur with Reviewer 4f5t's evaluation that the claims are not supported by sufficient evidence and agree with his evaluation:

> While the studies of 2023 unlearning challenge submissions are interesting, the paper would benefit from more rigorous and standardized evaluation. I do applaud the authors for including additional results in the rebuttal phase, but the results should be in the main body of the paper.
>
> Furthermore, the exact motivation of the paper is unclear as the research question raised in the abstract is not sufficiently supported by experiments. The results demonstrate computational savings (from reducing the size of unlearn set), but do not provide sufficient evidence regarding unlearning efficacy.
>
> Finally, the paper's title is also somewhat misleading, the proposed method does not "leverage" low influence points if they are simply disregarded, and stating that "unlearning is free" is also somewhat hard-to-understand.
>
> The conclusion is that the manuscript would benefit from a careful revision to improve clarity and / or precision.

While the paper consistently refers to NeurIPS 2023 challenge as a basis for selected evaluations, the field has developed since and this should be taken into account. In particular, the authors should in addition to above comments address the challenge raised by the paper
J. Hayes, I. Shumailov, E. Triantafillou, A. Khalifa and N. Papernot, "Inexact Unlearning Needs More Careful Evaluations to Avoid a False Sense of Privacy," in 2025 IEEE Conference on Secure and Trustworthy Machine Learning (SaTML), Copenhagen, Denmark, 2025, pp. 497-519, doi: 10.1109/SaTML64287.2025.00034.

**Resubmission Of Major Revision:**

The authors may consider submitting a major revision at a later time.